# Airway epithelial CD47 plays a critical role in inducing influenza virus-mediated bacterial super-infection

Sungmin Moon [1,2], Seunghan Han [1,2], In-Hwan Jang[3], Jaechan Ryu [4], Min-Seok Rha [5], Hyung-Ju Cho[5,6], Sang Sun Yoon[7], Ki Taek Nam [1], Chang-Hoon Kim[5,6], Man-Seong Park[8], Je Kyung Seong [9,10], Won-Jae Lee [3], Joo-Heon Yoon [5,6], Youn Wook Chung [1,6] ✉ & Ji-Hwan Ryu [1,2] ✉

Respiratory viral infection increases host susceptibility to secondary bacterial infections, yet the precise dynamics within airway epithelia remain elusive. Here, we elucidate the pivotal role of CD47 in the airway epithelium during bacterial super-infection. We demonstrated that upon influenza virus infection, CD47 expression was upregulated and localized on the apical surface of ciliated cells within primary human nasal or bronchial epithelial cells. This induced CD47 exposure provided attachment sites for *Staphylococcus aureus*, thereby compromising the epithelial barrier integrity. Through bacterial adhesion assays and in vitro pull-down assays, we identified fibronectin-binding proteins (FnBP) of *S. aureus* as a key component that binds to CD47. Furthermore, we found that ciliated cell-specific CD47 deficiency or neutralizing antibody-mediated CD47 inactivation enhanced in vivo survival rates. These findings suggest that interfering with the interaction between airway epithelial CD47 and pathogenic bacterial FnBP holds promise for alleviating the adverse effects of super-infection.

The 1918 influenza (H1N1) pandemic claimed the lives of more than 50 million people worldwide[1]. Most deaths were due to secondary bacterial pneumonia caused by common upper respiratory tract bacteria, such as *Staphylococcus aureus* and *Streptococcus pneumoniae*. Similarly, the illness of patients in the intensive care unit, infected with 2009 influenza A (H1N1), was exacerbated by secondary bacterial infections, with high morbidity and mortality rates[2,3]. The development and extensive use of antibiotics have played a crucial role in

reducing the severity of secondary bacterial infections. However, the recent emergence of antibiotic resistance has introduced a significant level of complexity to the treatment of these secondary bacterial infections[4]. Consequently, there is an urgent need to investigate and uncover the elusive molecular mechanisms responsible for secondary bacterial infections triggered by influenza.

Airway epithelial cells establish apical junctional complexes between neighboring cells to serve as a protective barrier against the

[1]Department of Biomedical Sciences, Yonsei University College of Medicine, Seoul 03722, Republic of Korea. [2]Brain Korea 21 PLUS Project for Medical Science, Yonsei University College of Medicine, Seoul 03722, Republic of Korea. [3]National Creative Research Initiative Center for Hologenomics and School of Biological Sciences, Seoul National University, Seoul 08826, Republic of Korea. [4]Microenvironment and Immunity Unit, Institut Pasteur, INSERM U1224 Paris, France. [5]Department of Otorhinolaryngology, Yonsei University College of Medicine, Seoul 03722, Republic of Korea. [6]Airway Mucus Institute, Yonsei University College of Medicine, Seoul 03722, Republic of Korea. [7]Department of Microbiology and Immunology, Yonsei University College of Medicine, Seoul 03722, Republic of Korea. [8]Department of Microbiology, Institute for Viral Diseases, Vaccine Innovation Center, Korea University College of Medicine, Seoul 02841, Republic of Korea. [9]Korea Mouse Phenotyping Center, Seoul National University, Seoul 08826, Republic of Korea. [10]Laboratory of Developmental Biology and Genomics, College of Veterinary Medicine, Seoul National University, Seoul 08826, Republic of Korea. ✉e-mail: chungyw@yuhs.ac; yjh@yuhs.ac

external environment. Apical junctional complexes consist of apical tight junctions and underlying adherens junctions, which facilitate cell–cell adhesion and maintain barrier integrity. Zonula occludens (ZO, e.g., ZO-1) and catenin (e.g., β-catenin) proteins connect the intracellular domains of tight junctions (e.g., Claudins and Occludin) and adherens junctions (e.g., E-cadherin) with cytoskeletal components, forming "cytosolic plaques"[5,6]. Tight junctions play a critical role in regulating the passage of ions and solutes through the paracellular space, effectively blocking the translocation of pathogens from the lumen to the interstitium. Consequently, viral infections that disrupt tight junctions can facilitate the translocation of pathogens and receptor exposure[7–10]. Thus, preserving the integrity of this barrier is crucial in preventing bacterial infection.

Respiratory viral infection increases host susceptibility to bacterial pathogens by *i)* interfering with antibacterial innate immune responses via interferon (IFN) induction[11–17] and depletion of alveolar macrophages[18]; or *ii)* providing binding sites for bacteria, including cellular receptors, such as intercellular adhesion molecule-1 (ICAM-1)[19–23], carcinogenic embryonic adhesion molecule 1 (CEACAM1)[20,24], platelet-activating factor receptor (PAF-r)[20,24], or extracellular matrix (ECM) proteins, such as fibronectin (FN)[24,25]. Given that PAF-r binds to the phosphorylcholine on *S. pneumoniae*'s cell wall, it has been proposed as a potential therapeutic target for secondary bacterial infections[26,27]. However, the use of a PAF-r antagonist did not demonstrate any effect in the secondary bacterial infection model[28]. As a result, it is essential to explore intervention strategies that specifically focus on disrupting the interaction between bacteria and cell receptors, utilizing antibody-mediated blockade, in treatment approaches.

CD47, also known as integrin-associated protein (IAP), is a widely expressed transmembrane glycoprotein. It serves as a "don't-eat-me" signal by interacting with the inhibitory receptor signal-regulatory protein alpha (SIRPα) on myeloid immune cells, thereby inhibiting phagocytosis of CD47-expressing erythrocyte[29]. Cancer and viral-infected hematopoietic cells overexpress CD47 for immune evasion[30,31]. Meanwhile, in non-hematopoietic cells, CD47 plays a role in tissue repair, contributing to improved healing and survival in various models such as skin thermal injury[32], organ transplant[33–37], and intestinal mucosal injury[38]. Notably, the specific involvement of CD47 in airway epithelium during super-infection remains unexplored. In a proteomics analysis of nasal epithelial cells infected with influenza virus, we observed the presence of CD47. Through the validation process, we uncovered that CD47 induced by viral infection was not only detected in the colonization site (nasal epithelium) but also at the site of infection (bronchial epithelium). Given that CD47 seems to be predominantly expressed in FoxJ1⁺ cells (deuterosomal cells and, to a lesser extent, multi-ciliated cells) based on single-cell RNA sequencing and immunostaining analysis of upper and lower airway primary epithelial cells, we initiated an investigation into the involvement of CD47 in secondary bacterial infections using a FoxJ1⁺ cell-specific CD47 gene-deletion mouse model and CD47 neutralizing antibodies. In our pursuit to identify a bacterial component that interact with CD47, we conducted experiments involving five mutant strains of *S. aureus* with deleted cell wall-anchored proteins. Remarkably, our investigations revealed that only fibronectin-binding proteins (FnBP) exhibited a strong affinity for CD47. To further solidify our findings, we employed a combination of FoxJ1⁺ cell-specific CD47 gene-deletion mice and a FnBP mutant strain of *S. aureus* in the context of super-infection. This approach allowed us to establish that the specific interaction between airway epithelial CD47 and bacterial FnBP plays a pivotal role in causing super-infection. By comprehensively exploring the role of CD47 in facilitating secondary bacterial infections, this study may pave the way for the development of innovative therapeutic approaches, ultimately leading to improved clinical outcomes for patients.

## Results

### CD47 is upregulated and localized to the apical surface of ciliated cells following infection of HBECs and HNECs with influenza virus

Limited research has investigated the localized surface remodeling of primary human nasal epithelial cells (HNECs) in the context of viral infection[10,23,24,39]. Consequently, the complete spectrum of upregulated adhesion molecules during respiratory viral infections that facilitate bacterial attachment and invasion remains unknown. To address this gap, the present study provides a comprehensive examination of HNEC responses to influenza virus pH1N1 infection. This investigation employs global proteome profiling of uninfected (Mock, $n = 4$) and influenza-infected (Virus-only, $n = 4$) HNECs utilizing isobaric tags for relative and absolute quantitation (iTRAQ) technique. A total of 3583 proteins were detected, 89 of which were significantly increased (52 proteins) or decreased (37 proteins) in the virus-infected HNECs compared to mock-infected controls ($p < 0.05$) (Supplementary Fig. 1a, b). Kyoto Encyclopedia of Genes and Genomes (KEGG) pathway analysis revealed four significantly enriched virus-related pathways in the virus-infected HNEC proteome, including "Influenza A" (Supplementary Fig. 1c). At three days post-infection (dpi), viral gene expression was elevated, accompanied by an increase in host innate immune responses, particularly type-I and type-III interferons (IFNs) (Supplementary Fig. 1d), while type-II IFN remained unchanged. Following viral infection, two candidate proteins displayed increased peptide expression and were expected to localize in the cellular membrane: leukocyte surface antigen CD47 (Q08722, FC = 1.15, $p = 0.001$), and carcinoembryonic antigen-related cell adhesion molecule 5 (CEACAM5, P06731, FC = 1.14, $p = 0.028$) (Supplementary Fig. 1b). Among these candidates, CD47 was selected for further investigation, and its expression was validated in both HNECs (Supplementary Fig. 2) and primary human bronchial epithelial cells (HBECs) (Fig. 1). Subsequent to viral infection, ZO-1 abundance decreased, and its cellular connections were disrupted, while CD47 exhibited increased expression in both HNECs (Supplementary Fig. 2a–c) and HBECs (Fig. 1a–c). In line with a prior study[38], functional annotation using gene ontology (GO) analysis demonstrated that CD47's biological processes were linked to *response to stimulus-related* GOs, *cell migration-related* GOs, *immune system process-related* GOs, cell surface receptor signaling pathway, and cellular component organization (Supplementary Fig. 1e).

Given that CD47 is transcriptionally regulated by nuclear factor kappa B (NF-κB) in a hepatocellular carcinoma mouse model[40], we examined whether airway epithelial CD47 is similarly regulated by the NF-κB pathway. In HBECs, we confirmed the activation of the NF-κB pathway induced by viral infection, evidenced by p65 phosphorylation at 1 h post-infection (hpi), along with the induction of the downstream target protein, intercellular adhesion molecule-1 (ICAM-1)[19,23]. CD47 protein abundance exhibited a slight increase after 1 hpi and reached its peak at 12 hpi (Supplementary Fig. 3a). Pre-treatment of HBECs with caffeic acid phenethyl ester (CAPE), a specific inhibitor of NF-κB, resulted in the inhibition of viral infection-induced p65 phosphorylation at 1 hpi (Supplementary Fig. 3b). Additionally, the upregulation of ICAM-1 and CD47 at 24 hpi was blocked (Fig. 1d), confirming the involvement of the NF-κB pathway in mediating CD47 upregulation in airway epithelial cells during viral infection. We also conducted an extensive investigation to determine whether CD47 induction is dependent on IFN, the production of which is NF-κB-dependent[41]. We observed an increase in CD47 expression following viral infection (Supplementary Fig. 3c, d), and this effect was replicated in HBECs treated with recombinant human IFN-β (Supplementary Fig. 3e) and IFN-λ1 (Supplementary Fig. 3f). Notably, the induction of CD47 by influenza virus was entirely suppressed when anti-IFNAR (Supplementary Fig. 3c) or anti-IFNLR (Supplementary Fig. 3d) neutralizing antibodies were applied. Additionally, the induction of CD47 by recombinant human IFN-β was effectively inhibited by anti-IFNAR

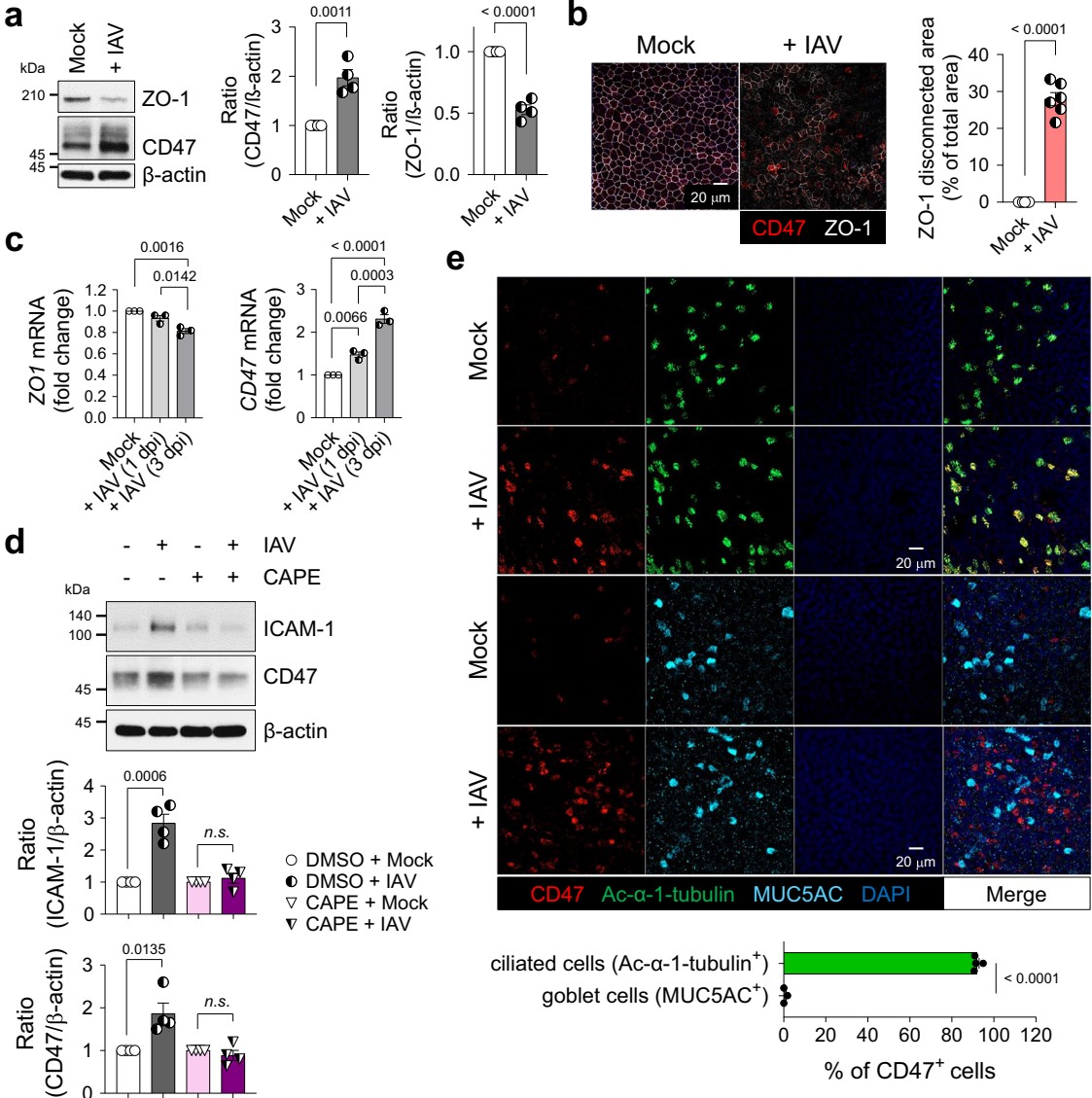

**Fig. 1 | Influenza virus induces CD47 expression on the apical surface of ciliated cells in an NF-κB-dependent manner.** HBECs were infected with (+ *IAV*) or without (*Mock*) influenza A virus. **a** Immunoblot analysis of junction protein ZO-1 and surface protein CD47 at 1 day post-infection (dpi). Normalized CD47 and ZO-1 protein levels are presented as bar graphs (Mock, $n = 4$; + IAV, $n = 4$). **b** Representative whole-mount images of ZO-1 (white) and CD47 (red) at 1 dpi. The area where ZO-1 disconnection occurred are presented as bar graphs (Mock, $n = 6$; + IAV, $n = 6$). **c** Quantitative PCR (qPCR) analysis of *ZO-1* and *CD47* mRNAs at 1 and 3 dpi [Mock, $n = 3$; + IAV (1 dpi), $n = 3$; + IAV (3 dpi), $n = 3$]. **d** HBECs were treated with or without 10 μM NF-κB inhibitor caffeic acid phenethyl ester (CAPE) for 1 h before influenza virus infection. Immunoblot analysis of ICAM-1 and CD47 at 1 dpi. Normalized ICAM-1 and CD47 protein levels are presented as bar graphs (DMSO + Mock, $n = 4$; DMSO + IAV, $n = 4$; CAPE + Mock, $n = 4$; CAPE + IAV, $n = 4$). **e** Whole-mount images of influenza virus-infected HBECs. Co-staining of CD47 (red) and ciliated cell-specific marker protein Ac-α-tubulin (green, $n = 4$) or goblet cell-specific marker protein MUC5AC (cyan, $n = 3$). Percentages of CD47-positive cells are presented as bar graphs. Data are presented as mean values ± SEM. Significance was determined by unpaired two-tailed Student's *t* test or one-way ANOVA with Tukey's multiple comparisons test. n.s. not significant. Source data are provided as a Source Data file.

neutralizing antibodies (Supplementary Fig. 3e), while CD47 induction by recombinant human IFN-λ1 was specifically blocked by anti-IFNLR neutralizing antibodies (Supplementary Fig. 3f). These results collectively demonstrate that CD47 induction occurs via the NF-κB/IFN pathway.

To ascertain if viral infection-induced CD47 expression is specific to certain cell types, we conducted co-immunostaining of airway epithelial cells and observed that CD47[+] HNECs (Supplementary Fig. 2d) and CD47[+] HBECs (Fig. 1e) were co-localized with the ciliated cell-specific marker, Ac-α-tubulin (92.0 ± 1.0% in HNECs; 91.1 ± 0.6% in HBECs) rather than the goblet cell-specific marker, MUC5AC (0.6 ± 0.6% in HNECs; 3.0 ± 1.6% in HBECs), during viral infection. Interestingly, co-immunostaining of CD47 and influenza nucleoprotein

(NP) revealed that not all CD47-positive cells completely overlapped with viral protein (Supplementary Fig. 3g). It's worth noting that ciliated cells are a part of the epithelium, lining the apical surface of the respiratory tract and exposing their cilia to the lumen[42]. Therefore, the detection of CD47 co-expressed with Ac-α-tubulin indicates that it is present on the apical side of the ciliated cells. In a similar context, the analysis of HNEC single-cell RNA sequencing (scRNA-seq) data revealed that *CD47*[+] clusters primarily consist of *FOXJ1*[+] cells, with a notable prevalence of deuterosomal cells and a partial presence of multi-ciliated cells (Supplementary Fig. 2e)[43]. These results reinforce a fundamental aspect of this study, underscoring the specific presence of CD47 expression in ciliated epithelial cells. Fluorescence-activated cell sorting (FACS) analysis further confirmed the presence of CD47

protein on the cell surface of HNECs, with increased expression observed during viral infection (Supplementary Fig. 2f). Therefore, these findings demonstrate that CD47 is induced and exposed on the apical surface of ciliated cells in respiratory epithelial cells infected with the influenza virus.

### Epithelial CD47 in HBECs and HNECs facilitates super-infection

To elucidate the mechanisms underlying susceptibility to secondary bacterial infection induced by viral infection, we developed an in vitro model of super-infection using HNECs and HBECs (Supplementary Fig. 4). HNECs simulate the upper airway, the initial site of viral infection, while HBECs represent the lower airway, where super-infection has a terminal impact. Following air–liquid interface (ALI) culture for 14 days (HNECs) or 21 days (HBECs), cells were infected with influenza virus pH1N1 and/or *S. aureus*. Since ZO-1 displayed reduced levels and disconnection during viral infection, we assessed the impact of secondary bacterial infection on HNECs and HBECs by evaluating paracellular permeability, changes in cellular morphology, and cell viability. When cells were co-infected with virus (MOI 1) and bacteria (MOI 5 for HNECs; MOI 3 for HBECs), paracellular permeability of fluorescein isothiocyanate (FITC)-dextran was significantly elevated (Supplementary Fig. 4a, e), and trans-epithelial electrical resistance (TEER) showed a substantial decrease (Supplementary Fig. 4b, f), as compared to mock-infected HNECs and HBECs. Notably, both of these parameters remained unchanged in cells subjected to either virus or bacteria single infection. Although no significant detrimental effect of super-infection was observed in lactate dehydrogenase (LDH) assay at 3 dpi (Supplementary Fig. 4c, g), visible cytopathogenic effects resulting from super-infection became apparent after 7 days in HNECs and 5 days in HBECs (Supplementary Fig. 4d, h). Given that HNECs exhibited greater resistance to super-infection compared to HBECs[44], it necessitated a longer incubation period with bacteria to observe a similar super-infection effect in HBECs. To investigate the function of CD47 in super-infection, we reduced CD47 expression by transfecting airway epithelial cells with lentiviral shRNA targeting the *CD47* gene. This resulted in a decrease in *CD47* transcript levels to $56.1 \pm 4.4\%$ in HNECs (Supplementary Fig. 5a) and $58.6 \pm 3.6\%$ in HBECs (Fig. 2a), accompanied by a corresponding reduction in CD47 protein levels, $41.7 \pm 5.2\%$ in HNECs (Supplementary Fig. 5b) and $54.2 \pm 6.3\%$ in HBECs (Fig. 2b). Subsequently, we assessed the impact of *CD47* knockdown on the response to super-infection by measuring paracellular FITC-dextran permeability and TEER at 3 dpi, which revealed a substantial attenuation of these effects in both HNECs (Supplementary Fig. 5c, d) and HBECs (Fig. 2c, d) following *CD47* shRNA treatment. Furthermore, we inhibited CD47 protein function by pre-incubating cells with α-hCD47 neutralizing antibodies before super-infection. This intervention resulted in significant mitigation of disruption in paracellular permeability and TEER at 3 dpi in HNECs (Supplementary Fig. 5e, f) and HBECs (Fig. 2c, d), as well as a reduction in the cytopathogenic effects of super-infection in HNECs at 7 dpi (Supplementary Fig. 5g) and HBECs at 5 dpi (Fig. 2e). In a super-infection model with *S. pneumoniae*, the treatment of *CD47* shRNA and α-hCD47 antibodies exhibited comparable protective effects on paracellular permeability and TEER in HBECs (Supplementary Fig. 6a, b). These results suggest that viral infection-induced epithelial CD47 may be exploited by pathogenic bacteria to induce super-infection by disrupting cellular junction integrity.

### Direct interactions between epithelial CD47 and the bacterial FnBP in viral-bacterial co-infected cells

With the exception of live *S. aureus*, no other factors, such as *S. aureus*-cultured media supernatant containing secreted proteins or extracellular vesicles (*S*), UV-killed *S. aureus* with intact structural proteins (*U*), or heat-killed *S. aureus* with denatured proteins (*H*), disrupted paracellular FITC-dextran permeability of HBECs infected with

influenza virus (Supplementary Fig. 7a, Fig. 3a, b). A whole-mount image of viral-infected HBECs briefly exposed to *S. aureus* (for 3 h, as longer exposure led to cell layer destruction) revealed the co-localization of bacteria with CD47 on the cell surface where ZO-1 disruption occurred (Fig. 3c). This observation led to our hypothesis that live *S. aureus* directly interacts with host CD47 to cause super-infection. To validate this hypothesis, we initially conducted a bacterial adhesion assay to quantify the number of bacteria adhering to HBECs after a 3-hour co-incubation. This analysis revealed a significant increase in the adherence of *S. aureus* to the cells following viral infection in HBECs (Fig. 3d). Additionally, the virus-induced bacterial adherence was inhibited when the cells were pre-incubated with α-hCD47 antibodies (Fig. 3d), suggesting that this cell-to-bacteria interaction may be dependent on CD47.

In our preliminary study, designed to elucidate the mechanistic basis how bacteria causing pneumonia following a secondary bacterial infection bind to CD47, we examined whether other pneumonia-causing bacteria bind to CD47. This was assessed by testing the adhesion of *S. aureus*, *S. pneumoniae* and *Pseudomonas aeruginosa* on A549 cells, with the commensal bacterium, *Staphylococcus epidermidis* serving as a negative control. The results showed that adherence of pneumonia-causing gram-positive bacteria *S. aureus* and *S. pneumoniae* to cellular CD47 was significantly inhibited following pre-incubation of the cells with α-hCD47 antibodies. However, adherence of pneumonia-causing gram-negative bacterium *P. aeruginosa*, was not affected by α-hCD47 antibodies. Moreover, the gram-positive commensal bacterium, *S. epidermidis*, did not bind to the cells (Supplementary Fig. 7b). Since only *S. aureus* and *S. pneumoniae* showed CD47-specific binding, we hypothesized that these two pathogens might share CD47-interacting components. Literature indicated that during infection, *S. aureus* and *S. pneumoniae* utilize structurally homologous adhesion molecules, such as Fn-binding protein (FnBP), to interact with integrin α5β1 by using Fn as a bridge. Specifically, FnBP A or B of *S. aureus* and pneumococcal adherence and virulence factor A (PavA) of *S. pneumoniae* are involved in these interactions[4,45,46].

To test whether FnBPs are required for *S. aureus* binding to CD47, we employed *S. aureus* laboratory strain 8325-4 (FnBP A+/B+) and three mutant strains (FnBP A+/B−, A−/B+, and A−/B−)[47]. In bacterial adhesion assay, we observed that the single deletion of FnBP A or B reduced *S. aureus* adherence by $23.7 \pm 1.7\%$ or $24.7 \pm 2.9\%$, respectively, while the double deletion of FnBP A and B synergically decreased adherence by $35.9 \pm 2.4\%$ (Fig. 3e). In contrast to the FnBP A+/B+ strain, where adherence to the viral infection-induced CD47 was inhibited by α-hCD47 antibodies in a concentration-dependent manner (Fig. 3f), the double deletion mutant strain (FnBP A−/B−) showed less adherence to viral infection-induced CD47, and this adherence was not affected by CD47 blocking (Fig. 3g). These findings indicated that FnBP A+/B+ binds to the surface of HBECs through CD47-specific ($26.4 \pm 1.9\%$) or non-specific interaction ($73.6 \pm 1.0\%$), whereas FnBP A−/B− only by non-specific interaction ($61.8 \pm 2.6\%$), suggesting a partial but direct interaction between CD47 and FnBP. To further support our hypothesis, we established an in vitro pull-down assay in which *S. aureus* was incubated with recombinant His-tagged hCD47 proteins and then pulled down using α-His-Dynabeads. Our results indicated a substantial increase in the precipitation of *S. aureus* in the pellets and a decreased presence in the supernatants when recombinant hCD47 was present, compared to the condition in which hCD47 was absent (Supplementary Fig. 7c), confirming that *S. aureus* directly binds to CD47. Lastly, to validate the direct interaction between CD47 and FnBP, we performed a pull-down assay using FnBP A+/B+ and FnBP A−/B−. Our results demonstrated that significantly less FnBP A−/B− ($15.8 \pm 6.9\%$) was precipitated in the pellets, even in the presence of recombinant hCD47, in contrast to FnBP A+/B+ ($83.2 \pm 5.7\%$) (Fig. 3h). By conducting in vitro pull-down assay using five mutant strains of *S. aureus*, in which cell wall-anchored proteins were deleted; *fnbB::Tn*

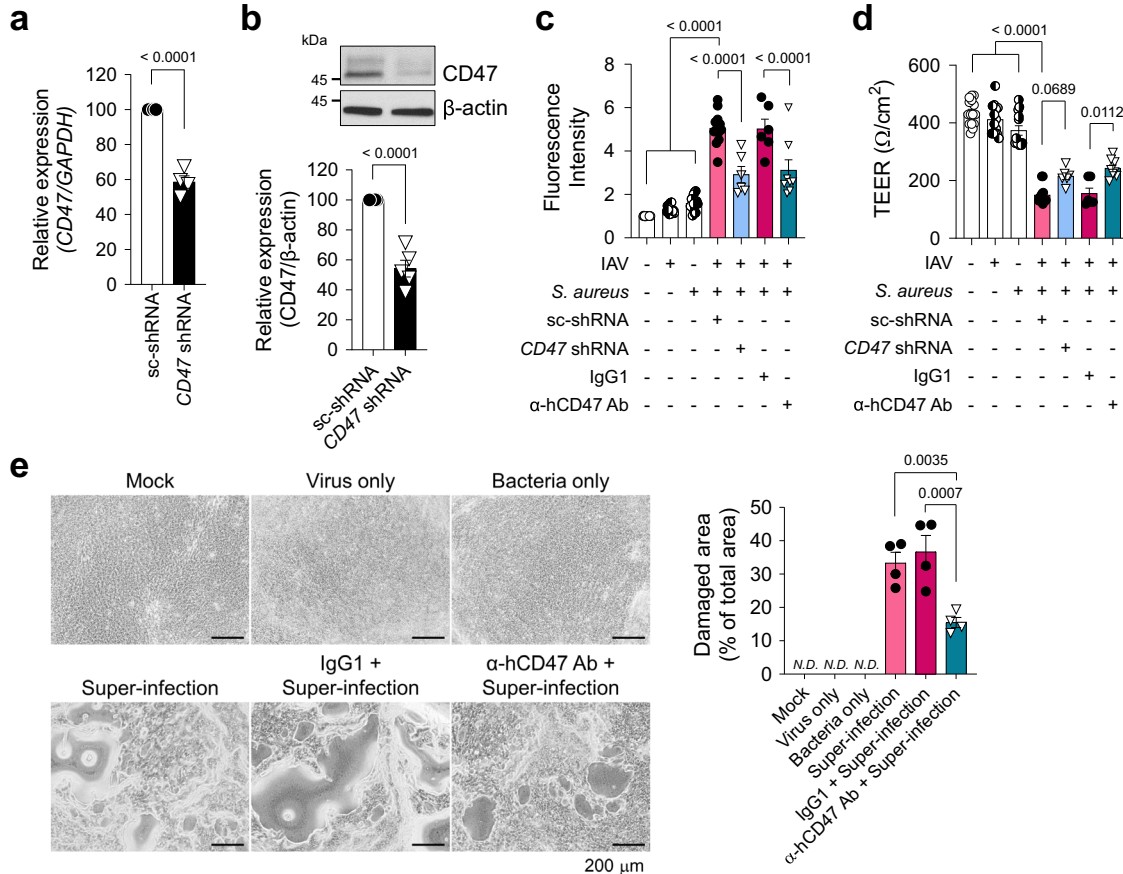

**Fig. 2 | Knock-down or neutralization of CD47 protects HBECs from viral–bacterial co-infection.** For *CD47* knock-down, HBECs were transfected with scrambled shRNA (*sc-shRNA*) or shRNA targeting *CD47* (*CD47 shRNA*) using a lentiviral delivery system. For CD47 neutralization, HBECs were treated with either IgG1 (MOPC-21) or α-hCD47 (B6H12.2) antibodies. **a** Gene expression of *CD47* was analyzed using qRT-PCR (normalized by *GAPDH* mRNA) (sc-shRNA, *n* = 4; *CD47* shRNA, *n* = 4). **b** Protein expression of CD47 was analyzed using immunoblotting (sc-shRNA, *n* = 5; *CD47* shRNA, *n* = 5). **c** Paracellular FITC-dextran permeability was measured in Mock (*n* = 16), Virus only (IAV, *n* = 16), Bacteria only (*S. aureus*, *n* = 16), sc-shRNA + Super-infection (*n* = 12), *CD47* shRNA + Super-infection (*n* = 6), IgG1 +

Super-infection (*n* = 6), and α-hCD47 antibodies (Ab) + Super-infection (*n* = 8). **d** Trans-epithelial resistance was measured in Mock (*n* = 15), IAV (*n* = 14), *S. aureus* (*n* = 13), sc-shRNA + Super-infection (*n* = 13), *CD47* shRNA + Super-infection (*n* = 6), IgG1 + Super-infection (*n* = 6), and α-hCD47 Ab + Super-infection (*n* = 8). **e** Microscopic images of HBECs at 5 dpi. The percentage of the damaged area is presented as bar graphs (Mock, *n* = 3; Virus only, *n* = 3; Bacteria only, *n* = 3; Super-infection, *n* = 4; IgG1 + Super-infection, *n* = 4; α-hCD47 Ab + Super-infection, *n* = 4). Data are presented as mean values ± SEM. Significance was determined by unpaired two-tailed Student's *t* test or one-way ANOVA with Tukey's multiple comparisons test. N.D. not determined. Source data are provided as a Source Data file.

(*ΔfnbB*), *sasG::Tn* (*ΔsasG*), *isdB::Tn* (*ΔisdB*), *sdrE::Tn* (*ΔsdrE*), and *clfA::Tn* (*ΔclfA*). We confirmed that only FnBP mediates the interaction between *S. aureus* and CD47 (Supplementary Fig. 7d, e). This observation supports the conclusion that *S. aureus* directly binds to HBECs through a specific interaction between FnBP and CD47.

### Foxj1<sup>Cre</sup>-specific disruption or neutralization of CD47 protects mice from super-infection

Considering the increased expression of CD47 in the tracheal and lung epithelium (Supplementary Fig. 8a–e) and the decreased expression of ZO-1 in the lung lysates of C57BL/6 WT mice following infection with influenza virus (Supplementary Fig. 8f), we subsequently investigated the pathophysiological role of viral infection-induced CD47. As CD47 is ubiquitously expressed, understanding the contributions of specific CD47-expressing cell types during super-infection is crucial for gaining detailed mechanistic insights. Given that epithelial CD47 was exclusively induced in ciliated cells during viral infection (Supplementary Fig. 2d, Fig. 1e), we generated mice with selective *CD47* loss in ciliated cells by creating *CD47* floxed mice (*CD47*[f/f]), bred to constitutively express Cre under the control of the Foxj1 promoter (*Foxj1*[Cre])[48]. Cre specificity was validated by crossing with tdTomato reporter mice (Supplementary Fig. 9a), in which *Foxj1*[tdTomato-]

expressing cells were co-stained with the ciliated cell-specific marker protein Ac-α-tubulin. Immunofluorescence confirmed the specific loss of *CD47* in ciliated cells of *CD47*[FoxJ1] mice (Supplementary Fig. 9b). To ensure that CD47 inhibition during viral infection does not enhance innate immunity nor induce quicker virus clearance[31], we also generated mice with selective loss of *CD47* in myeloid immune cells using *LysM*[Cre] mice (*CD47*[LysM]). Notably, *CD47*[FoxJ1] and *CD47*[LysM] mice did not exhibit differences in respiratory function under baseline conditions, compared with *CD47*[f/f] mice (Supplementary Fig. 9c).

To investigate the necessity of airway epithelial CD47 in the context of viral-bacterial co-infection, we established two in vivo mouse models of super-infection. As depicted in Supplementary Figs. 10a and 10k, we employed two different concentrations of virus and bacteria in both the *CD47* gene deletion experiment (100 PFU of virus and 1 ×10[8] CFU of bacteria) and the CD47 neutralizing experiment (10 PFU of virus and 5 ×10[5] CFU of bacteria). Four groups of mice, including an uninfected group (Mock), a viral infection group, a bacterial infection group, and a super-infection group, were daily monitored for alterations in body weight and mortality over a 29-day period (Supplementary Fig. 10b, c, l, m). Remarkably, approximately 98% of the mice succumbed to the super-infection within 10 days,

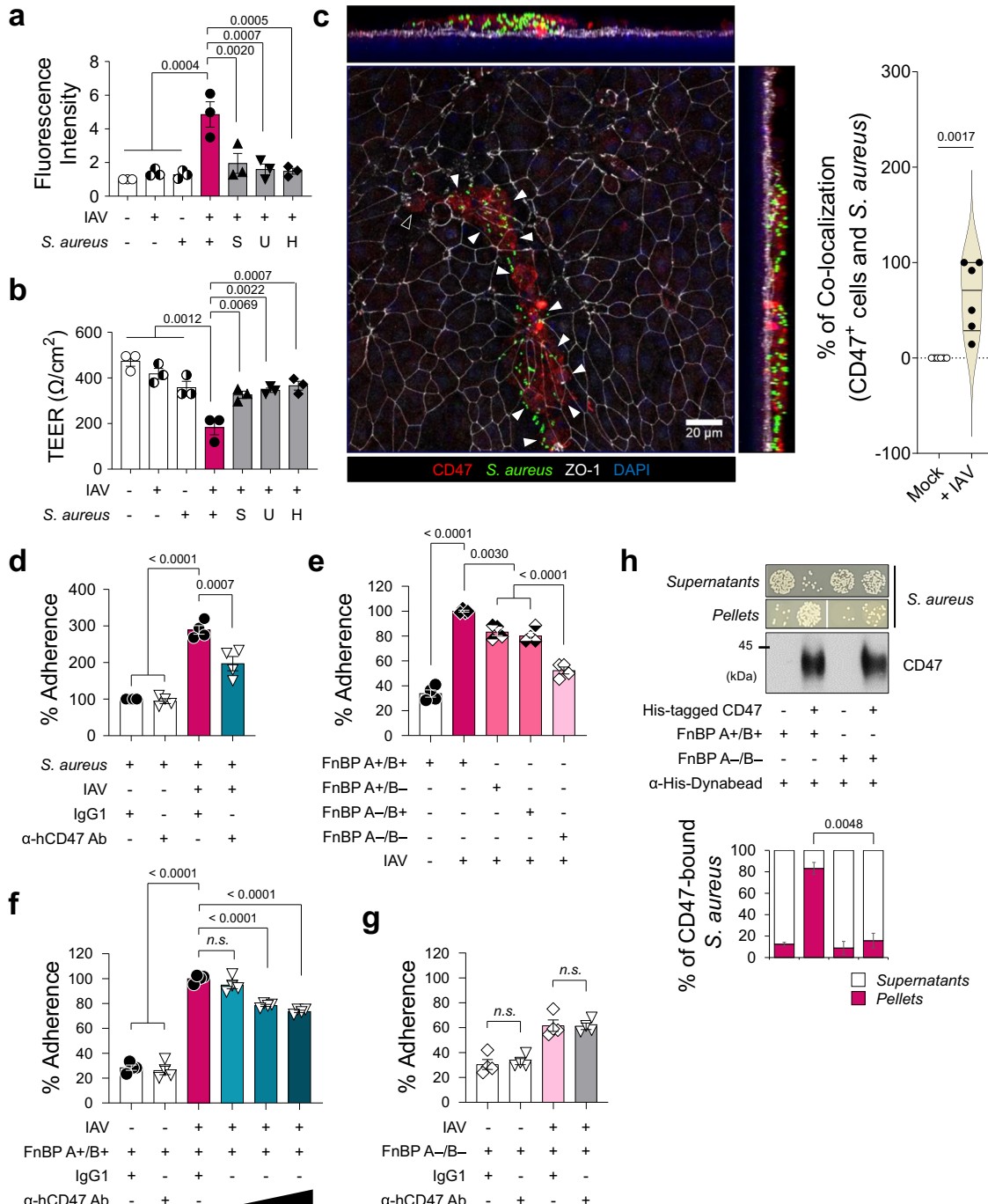

**Fig. 3 | Epithelial CD47 and bacterial FnBP directly interacts. a, b** Influenza virus-infected HBECs (*IAV*) infected with live *S. aureus* or *S. aureus*-derived samples (*S*, supernatant of *S. aureus*-cultured media; *U*, UV-killed *S. aureus*; *H*, heat-killed *S. aureus*). Paracellular FITC-dextran permeability (**a**) and trans-epithelial electrical resistance (**b**) of 7 groups: *i*) Mock (*n* = 3), *ii*) Virus only (*n* = 3), *iii*) Bacteria only (*n* = 3), *iv*) Super-infection (*n* = 3), *v*) Virus with *S* (*n* = 3), *vi*) Virus with *U* (*n* = 3), and *vii*) Virus with *H* (*n* = 3). **c** Whole mount image of CD47 (red) and *S. aureus* (green) in the influenza virus-infected HBECs at 1 dpi. An open arrow head indicates CD47[+] cell without *S. aureus* and closed arrow heads indicate CD47[+] cells with *S. aureus*. Co-localization of CD47[+] cells and *S. aureus* are presented as violin plots (Mock, *n* = 6; + IAV, *n* = 6). **d–g** Bacterial adhesion assay. Colonization of *S. aureus* was assessed in HBECs inoculated with the virus (MOI 1) for 2 h before treatment with IgG1 control antibodies or α-hCD47 neutralizing antibodies (2 h), followed by *S. aureus* (MOI 3) infection for 3 h (IgG1 + *S. aureus*, *n* = 4; α-hCD47 Ab + *S. aureus*, *n* = 4; IgG1 + Super-infection, *n* = 4; α-hCD47 Ab+ Super-infection, *n* = 4) (**d**). Adherence of *S. aureus* WT (FnBP A+/B+, *n* = 4) and three mutant strains (FnBP A+/B−, *n* = 4; FnBP A−/B+, *n* = 4;

FnBP A−/B−, *n* = 4) was assessed in HBECs inoculated with the virus (MOI 1) for 2 h, followed by *S. aureus* (MOI 3) infection for 3 h (**e**). Colonization of *S. aureus* WT (FnBP A+/B+) (**f**) and double deletion mutant (FnBP A−/B−) (**g**) was assessed in HBECs inoculated with the virus (MOI 1) for 2 h before treatment with IgG1 control antibodies (*n* = 4) or α-hCD47 neutralizing antibodies (1, 5, and 10 µg/mL for FnBP A+/B+, and 10 µg/mL for FnBP A−/B−, *n* = 4 each) for 2 h, followed by *S. aureus* (MOI 3) infection for 3 h. **h** Pull-down assay using His-tagged hCD47 recombinant protein. Bacterial plating [FnBP A+/B+ and FnBP A−/B− (*n* = 3 each in the absence or in the presence of His-tagged hCD47)] and immunoblot analysis were performed using supernatants and pellets after separation with α-His-Dynabeads™/Dyna-Mag™-2 system. The graphs present the percentage of colony numbers grown in the culture of the supernatants or the pellets. Data are presented as mean values ± SEM. Significance was determined by one-way ANOVA with Tukey's multiple comparisons test or unpaired two-tailed Student's *t* test. n.s. not significant. Source data are provided as a Source Data file.

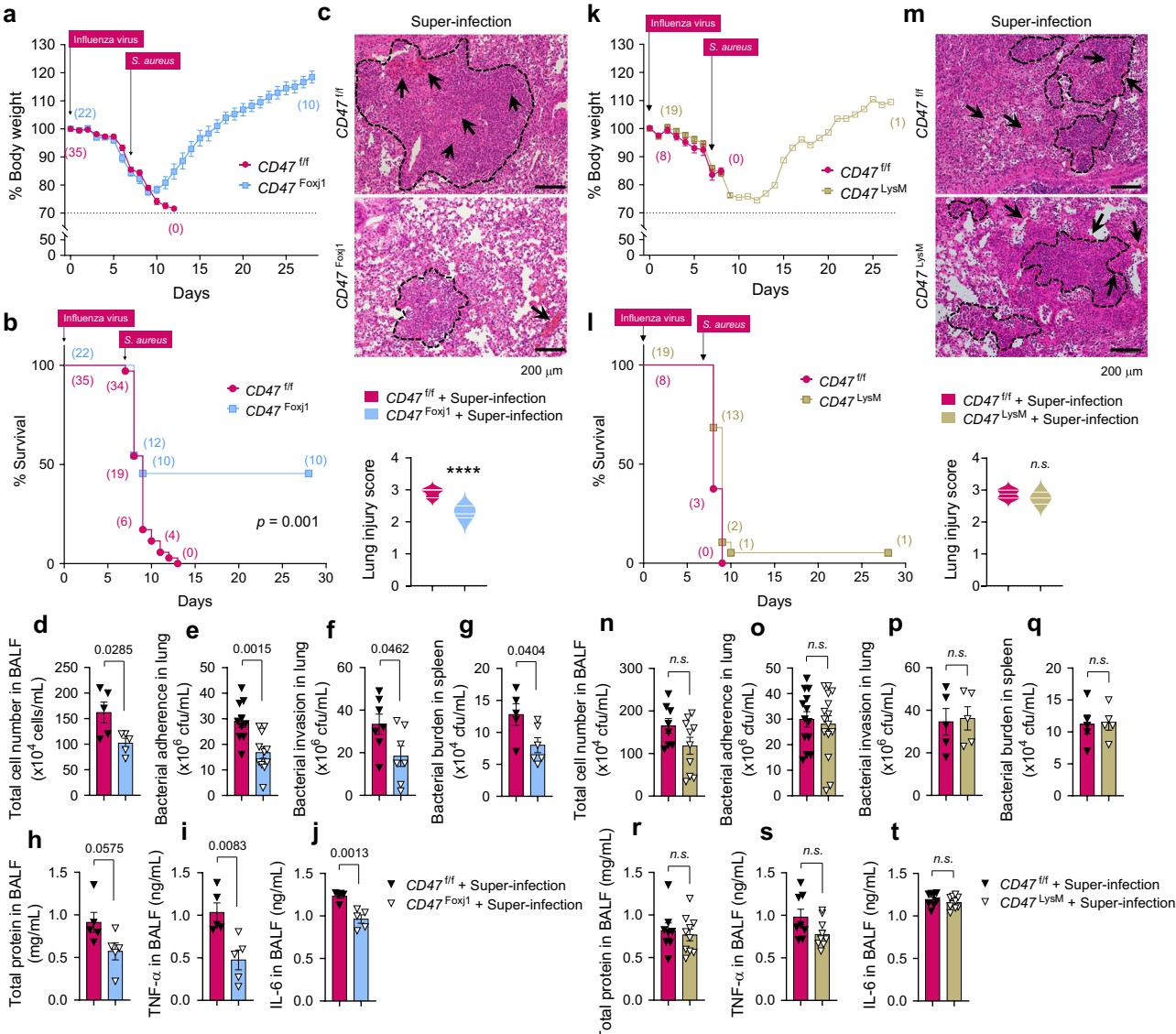

**Fig. 4 | Foxj1^Cre^-Specific, but not LysM^Cre^-Specific, *CD47* disruption protects mice from super-infection.** 6–8-weeks-old (18–21 g of body weight) *FoxJ1*-Cre;floxed (*CD47*^Foxj1^), *LysM*-Cre;floxed (*CD47*^LysM^), and control floxed (*CD47*^f/f^) mice were infected with 100 PFU of influenza virus on day 0, and 1 ×10^8^ CFU of *S. aureus* on day 7. **a**, **b**, **k**, **l** Body weight loss (**a**, **k**) and survival rates (**b**, **l**) were monitored for 29 days. The dotted line indicates the body weight exclusion cut-off. A mantel cox survival analysis was used to compare the survival rates between groups. The numbers in parenthesis represent the count of surviving mice. **c**, **m** Representative hematoxylin and eosin (H&E) staining of lung sections. The dotted lines indicate lymphocytic infiltration and arrows indicate alveolar hemorrhage. Lung injury scores are presented as violin plots in *CD47*^Foxj1^ mice (*CD47*^f/f^, $n = 8$; *CD47*^Foxj1^, $n = 5$) (**c**) and *CD47*^LysM^ mice (*CD47*^f/f^, $n = 8$; *CD47*^LysM^, $n = 8$) (**m**). **d**–**j**, **n**–**t** Tissue injury parameters were measured at 24 h after bacterial infection; total cell number in BAL fluids (BALF) of *CD47*^Foxj1^ mice (*CD47*^f/f^, $n = 5$; *CD47*^Foxj1^, $n = 5$) (**d**) and *CD47*^LysM^ mice

(*CD47*^f/f^, $n = 8$; *CD47*^LysM^, $n = 10$) (**n**); bacterial adherence in the lung of *CD47*^Foxj1^ mice (*CD47*^f/f^, $n = 10$; *CD47*^Foxj1^, $n = 11$) (**e**) and *CD47*^LysM^ mice (*CD47*^f/f^, $n = 13$; *CD47*^LysM^, $n = 15$) (**o**); bacterial invasion in the lung of *CD47*^Foxj1^ mice (*CD47*^f/f^, $n = 7$; *CD47*^Foxj1^, $n = 7$) (**f**) and *CD47*^LysM^ mice (*CD47*^f/f^, $n = 5$; *CD47*^LysM^, $n = 5$) (**p**); and bacterial burden in the spleen of *CD47*^Foxj1^ mice (*CD47*^f/f^, $n = 5$; *CD47*^Foxj1^, $n = 6$) (**g**) and *CD47*^LysM^ mice (*CD47*^f/f^, $n = 5$; *CD47*^LysM^, $n = 5$) (**q**); total protein concentrations in BALF of *CD47*^Foxj1^ mice (*CD47*^f/f^, $n = 5$; *CD47*^Foxj1^, $n = 5$) (**h**) and *CD47*^LysM^ mice (*CD47*^f/f^, $n = 8$; *CD47*^LysM^, $n = 10$) (**r**); and inflammatory cytokines TNF-α in BALF of *CD47*^Foxj1^ mice (*CD47*^f/f^, $n = 5$; *CD47*^Foxj1^, $n = 5$) (**i**) and *CD47*^LysM^ mice (*CD47*^f/f^, $n = 8$; *CD47*^LysM^, $n = 10$) (**s**), and IL-6 in BALF of *CD47*^Foxj1^ mice (*CD47*^f/f^, $n = 5$; *CD47*^Foxj1^, $n = 5$) (**j**) and *CD47*^LysM^ mice (*CD47*^f/f^, $n = 8$; *CD47*^LysM^, $n = 10$) (**t**). Data are presented as mean values ± SEM. Significance was determined by unpaired two-tailed Student's *t* test. Source data are provided as a Source Data file.

experiencing unrecoverable weight loss. To understand the underlying causes of this lethality, we assessed *i)* histological damage scores of lung sections, *ii)* total cell numbers in bronchoalveolar lavage fluids (BALF), *iii)* bacterial burden in the lung and the spleen, *vi)* concentrations of total protein, and *v)* levels of pro-inflammatory cytokines (TNF-α and IL-6) in BAL fluids at 24 h following bacterial infection. In all these parameters, the super-infection group exhibited significantly increased values compared to the mock, viral infection, and bacterial infection groups (Supplementary Fig. 10e–j).

Subsequently, we investigated whether the presence of airway epithelial CD47 or myeloid CD47 is essential for super-infection. We conducted experiments using 6–8-weeks-old mice weighing between 18–21 grams with specific genetic modifications: FoxJ1-Cre;floxed (*Cd47*^Foxj1^), LysM-Cre;floxed (*Cd47*^LysM^), and control floxed (*Cd47*^f/f^) mice. These mice were infected with 100 PFU of influenza virus on day 0, followed by an infection of 1 ×10^8^ CFU of *S. aureus* on day 7. Body weight loss and survival rates were monitored for 29 days. Notably, *CD47*^Foxj1^ mice exhibited a recovery in body weight loss and improved survival rates compared to *CD47*^f/f^ mice (Fig. 4a, b), while *CD47*^LysM^ mice

did not show the same improvements (Fig. 4k, l). Moreover, signs of pneumonia were significantly alleviated in *CD47*[FoxJI] mice (Fig. 4d–j) but not in *CD47*[LysM] mice (Fig. 4n–t) when compared to *CD47*[f/f] mice. This improvement was evident through several measures: *i)* a reduced histological lung injury score (Fig. 4c), *ii)* decreased total cell numbers in BALF (Fig. 4d), *iii)* diminished bacterial adherence and invasion in the lung (Fig. 4e, f, respectively) and bacterial burden in the spleen (Fig. 4g), *iv)* reduced total protein levels (Fig. 4h), and *v)* decreased levels of TNF-α and IL-6 in BALF 24 h after bacterial infection (Fig. 4i, j, respectively). These findings underscore the protective effect of inhibiting CD47 specifically in ciliated cells rather than myeloid cells against super-infection.

Next, we examined the efficacy of antibody-mediated CD47 blockade in inhibiting super-infection. C57BL/6 WT mice pre-treated with α-mCD47 neutralizing antibodies exhibited a recovery in body weight loss (Fig. 5a) and improved survival rates (Fig. 5b) compared to C57BL/6 WT mice pre-treated with IgG2a control antibodies. Additionally, signs of pneumonia were significantly mitigated in α-mCD47-treated mice compared to IgG2a-treated mice, as evidenced by a decrease in histological lung injury score (Fig. 5c), a reduction in total cell number in BALF (Fig. 5d), a decrease in bacterial adherence and invasion in the lung (Fig. 5e, f, respectively) and bacterial burden in the spleen (Fig. 5g), a decline in total protein levels (Fig. 5h), and a decrease in TNF-α and IL-6 in BALF (Fig. 5i, j). These findings indicate that therapeutically inhibiting CD47 protects mice from super-infection.

To fully confirm the interaction between FnBP and CD47 in vivo, we infected both *CD47*[f/f] and *CD47*[FoxJI] mice with two *S. aureus* strains, FnBP A+/B+ and FnBP A−/B−, to verify whether the observed phenotype aligns with our expected model[49]. As expected, FnBP A−/B− exhibited reduced pathogenicity when infecting *CD47*[f/f] mice, similar to the effect of FnBP A+/B+ in *CD47*[FoxJI] mice, while there was no observable effect when either FnBP A+/B+ or FnBP A−/B− infected *CD47*[FoxJI] mice (Fig. 6a–j). Taken together, our results demonstrate that both airway epithelial CD47 and *S. aureus* FnBP are essential for the development of super-infection (Fig. 6k).

## Discussion

The upregulation of CD47 in response to viral infections other than influenza has been documented in previous studies[50–52]. For instance, human respiratory syncytial virus (RSV) and human parainfluenza virus 3 (HPIV3) infections have been shown to increase CD47 levels[50]. Additionally, CD47 expression is elevated in cells infected with SARS-CoV-2[52]. McLaughlin et al. conducted an analysis of publicly available proteomics[53] and transcriptomics[54] data, revealing increased CD47 expression in SARS-CoV-2-infected HBECs and Caco2 cells. However, the specific role of viral infection-induced CD47 in the context of secondary bacterial infection remains unclear. In this study, we provide evidence that: *i)* Influenza virus infection induces the upregulation of airway epithelial CD47, particularly in ciliated cells, accompanied by ZO-1 disruption. *ii)* FnBP, a member of the microbial surface components recognizing adhesive matrix molecules (MSCRAMM) family, plays a crucial role in the attachment of pneumonia-causing pathogenic bacteria such as *S. aureus*. *iii)* CD47 neutralizing antibodies protect mice from secondary bacterial infection by blocking the direct interaction between CD47 and FnBP. These findings elucidate the intricate interplay between viral infections and bacterial pathogens, particularly regarding CD47 upregulation and its implications for host defenses.

The upper and lower airways are lined by a pseudostratified epithelium composed of ciliated, goblet, and basal cells[42]. In human tissues, influenza viruses primarily bind to ciliated cells, whereas in differentiated airway epithelial cell cultures, they infect non-ciliated cells and, to a lesser extent, ciliated cells[55]. Recent studies have shown that SARS-CoV-2 also exhibit a preference for targeting ciliated cells and interferes with mucociliary clearance[56]. It's important to note that

while CD47 induction by viral infection is specific to ciliated cells (Fig. 1e and Supplementary Fig. 2d), not all ciliated cells were infected (Supplementary Fig. 3g). Given that virus-induced IFNs are secreted and signal on neighboring cells[57], the observed expression pattern can be explained. Although it could be speculated that virus-infected cells experience a more suppressed IFN response due to viral mechanisms inhibiting IFN signals, whereas uninfected cells exhibit a full response to IFN, resulting in the upregulation of CD47, the precise mechanism by which neighboring infected cells induce CD47 in ciliated cells remains unclear at this stage.

Pathogenic bacteria possess a multitude of virulence factors that enable them to bind to host receptors or extracellular matrix (ECM) components[4,45,46]. While our in-depth investigation primarily focused on the membrane proteins of *S. aureus* as a microbial component responsible for interacting with epithelial CD47, additional research is necessary to uncover the specific surface proteins binding to CD47 in other pneumonia-causing pathogenic bacteria, such as *S. pneumoniae, Haemophilus influenzae, and Streptococcus hemolyticus*. The FnBP-Fn-α5β1 integrin pathway is recognized as the primary adhesion and internalization process; however, it is important to note that additional factors have been shown to influence the efficiency of this process. For example, Dziewanowska et al. reported that the direct interaction between FnBPs and Hsp60 maximizes the internalization efficiency in epithelial cells[58]. Further investigation is required to determine whether CD47 acts as a co-receptor, enhancing the binding of FnBP-Fn to α5β1 integrin.

We assessed CD47 mRNA levels (Supplementary Fig. 8b) and protein levels (Supplementary Fig. 8c, d) at multiple time points, including 2, 3, 5, 7, and 10 dpi. Notably, both mRNA and protein levels exhibited a gradual increase, reaching their peak at 5 dpi, followed by a subsequent decrease. Intriguingly, in contrast to this observation, *S. aureus* single infection failed to induce CD47 expression in both in vitro and in vivo settings. Furthermore, no synergistic effect was observed with viral infection, as shown in Supplementary Fig. 8g, h. This time course data provides valuable insights into how CD47 expression correlates with the window of enhanced susceptibility to secondary infection, which is typically observed between days 5–10 post-infection with influenza virus[59,60]. Two potential explanations may account for the expression of CD47 on the surface of host epithelial cells in response to viral infection. In the context of viral propagation, viruses may employ CD47 surface expression as a strategy to evade immune cells presenting "don't-eat-me" signals, similar to their actions in infected hematopoietic cells[30,31]. Second, in order to support their systemic proliferation, viruses could induce epithelial cells to express binding sites for bacteria. In the context of host signaling, it is conceivable that virus-infected epithelial cells might stimulate CD47 expression via the NF-κB signaling pathway, as has been reported for ICAM-1[19,23].

Due to the methodological challenges inherent in our primary epithelial cell ALI culture setup, we were unable to distinctly differentiate between bacterial adherence and invasion in vitro. Instead, we utilized a super-infection mouse model to evaluate both the adherence of *S. aureus* and its invasion into the lung tissue[61], as well as bacterial CFU in the spleen as an indicator of systemic bacterial dissemination[62]. In order to establish an in vivo correlation with the observed loss of the epithelial barrier function in our in vitro super-infection model (Supplementary Fig. 5 and Fig. 2), we validated the downregulation of ZO-1 in the lung following viral infection (Supplementary Fig. 8f) and the increases in *S. aureus* invasion in the lung (Supplementary Fig. 10g and Fig. 5f) and bacteremia during super-infection (Supplementary Fig. 10h and Fig. 5g).

Our findings demonstrated that the absence of CD47 in ciliated cells (Fig. 4f, g) or the absence of FnBP in *S. aureus* (Fig. 6f, g) indeed provided protection against *S. aureus* invasion. Notably, this protective effect was not observed in myeloid cell-specific knockout mice

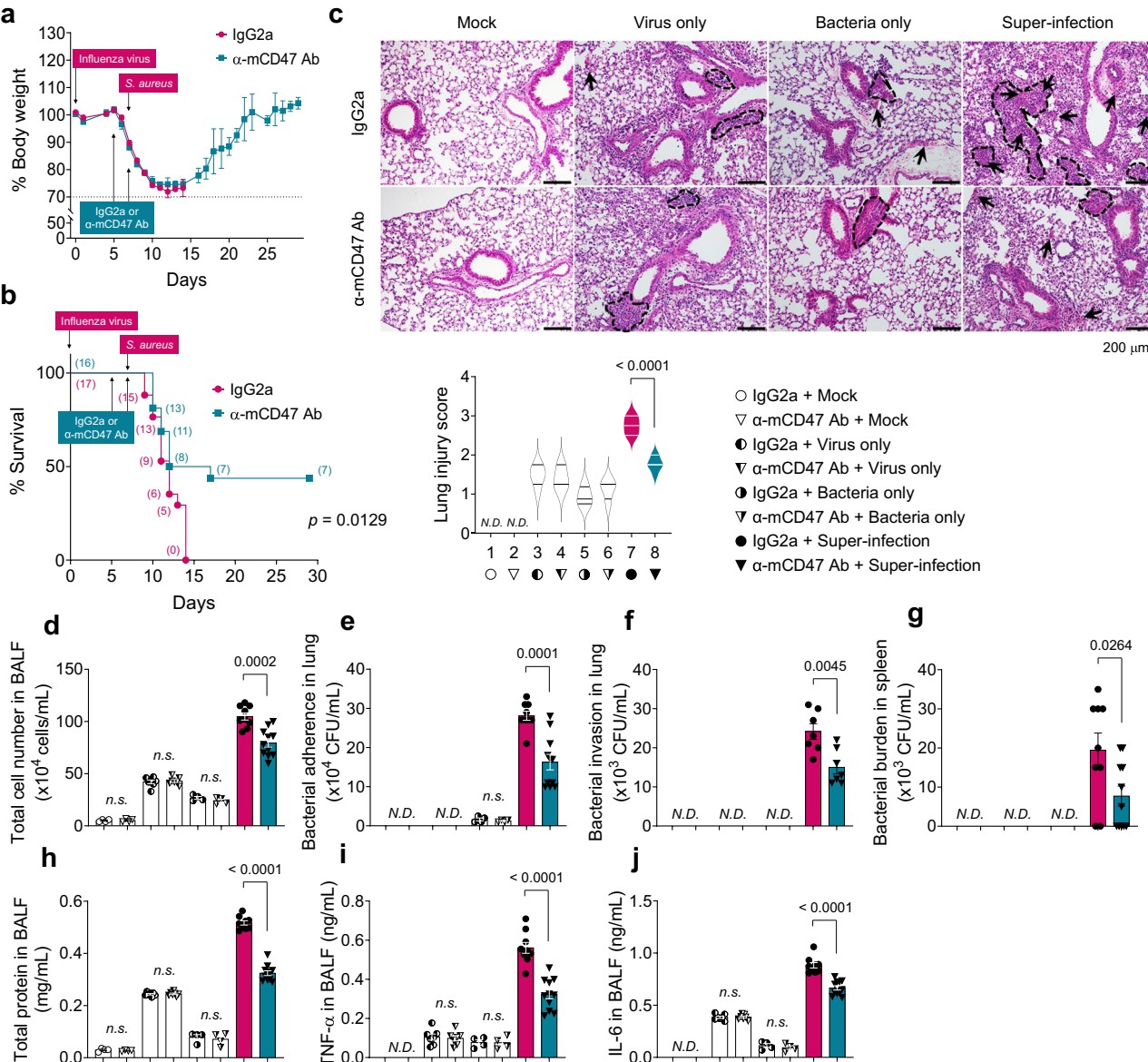

**Fig. 5 | CD47 neutralization enhances protection against super-infection.** For neutralization test, 6–8-weeks-old (18–21 g of body weight) C57BL/6 WT mice were infected with 10 PFU of influenza virus on day 0 and $5 \times 10^5$ CFU of *S. aureus* on day 7. Before bacterial infection, mice were intranasally treated twice at day 5 and 7 with IgG2a control antibodies (2A3, *n* = 17) or α-mCD47 neutralizing antibodies (MIAP301, *n* = 16). **a, b** Body weight loss (**a**) and survival rates (**b**) were monitored for 29 days. The dotted line indicates the body weight exclusion cut-off. A mantel cox survival analysis was used to compare the survival rates between groups. The numbers in parenthesis are numbers of survived mice. **c** Representative hematoxylin and eosin (H&E) staining of lung sections (IgG2a + Mock, *n* = 4; α-mCD47 Ab + Mock, *n* = 4; IgG2a + Virus only, *n* = 7; α-mCD47 Ab + Virus only, *n* = 7; IgG2a + Bacteria only, *n* = 4; α-mCD47 Ab + Bacteria only, *n* = 4; IgG2a + Super-infection,

*n* = 9; α-mCD47 Ab + Super-infection, *n* = 11). The dotted lines indicate lymphocytic infiltration and arrows indicate alveolar hemorrhage. Lung injury scores are presented as violin plots. **d–j** Tissue injury parameters were measured at 24 h after bacterial infection (IgG2a + Mock, *n* = 4; α-mCD47 Ab + Mock, *n* = 4; IgG2a + Virus only, *n* = 7; α-mCD47 Ab + Virus only, *n* = 7; IgG2a + Bacteria only, *n* = 4; α-mCD47 Ab + Bacteria only, *n* = 4; IgG2a + Super-infection, *n* = 7-9; α-mCD47 Ab + Super-infection, *n* = 7-11); total cell number in BAL fluids (BALF) (**d**), bacterial adherence (**e**) and invasion (**f**) in the lung, and the bacterial burden in the spleen (**g**), total protein concentrations (**h**) in BALF, and inflammatory cytokines TNF-α (**i**) and IL-6 (**j**) in BALF. Data are presented as mean values ± SEM. Significance was determined by unpaired two-tailed *t* test. n.s. not significant. N.D. not determined. Source data are provided as a Source Data file.

(Fig. 4p, q). Additionally, the use of anti-CD47 neutralizing antibody treatment proved effective in preventing the bacterial invasion (Fig. 5f, g). Nonetheless, we have not yet elucidate the precise mechanisms by which accumulated bacteria breach the barrier more effectively. Further investigation into how *S. aureus* interacts with CD47 to facilitate its invasion process remains a valuable and ongoing area of research.

We conducted a comprehensive evaluation of innate immunity and continuously monitored viral loads in the lungs of IAV-infected mice up to 10 dpi. Our observations revealed that there were no significant

differences in viral titers between *CD47*[f/f], *CD47*[Foxj1], and *CD47*[LysM] mice at 2, 3, 4, and 7 dpi (Supplementary Fig. 9j, k), as well as in IFN-λ2/3 levels (Supplementary Fig. 9g) and viral loads (Supplementary Fig. 9h, i) between *CD47*[f/f] mice and *CD47*[Foxj1] mice at 10 dpi. These findings suggest that, at the time points examined, CD47 deficiency in ciliated cells or myeloid cells does not seem to have a substantial impact on innate immunity. A previous study has highlighted the crucial role of CD47 as an "eat me" signal critical for macrophage clearance of neutrophils that have ingested *S. aureus*, leading to increased infection in the absence of CD47[63]. While prior research has focused on CD47's involvement in

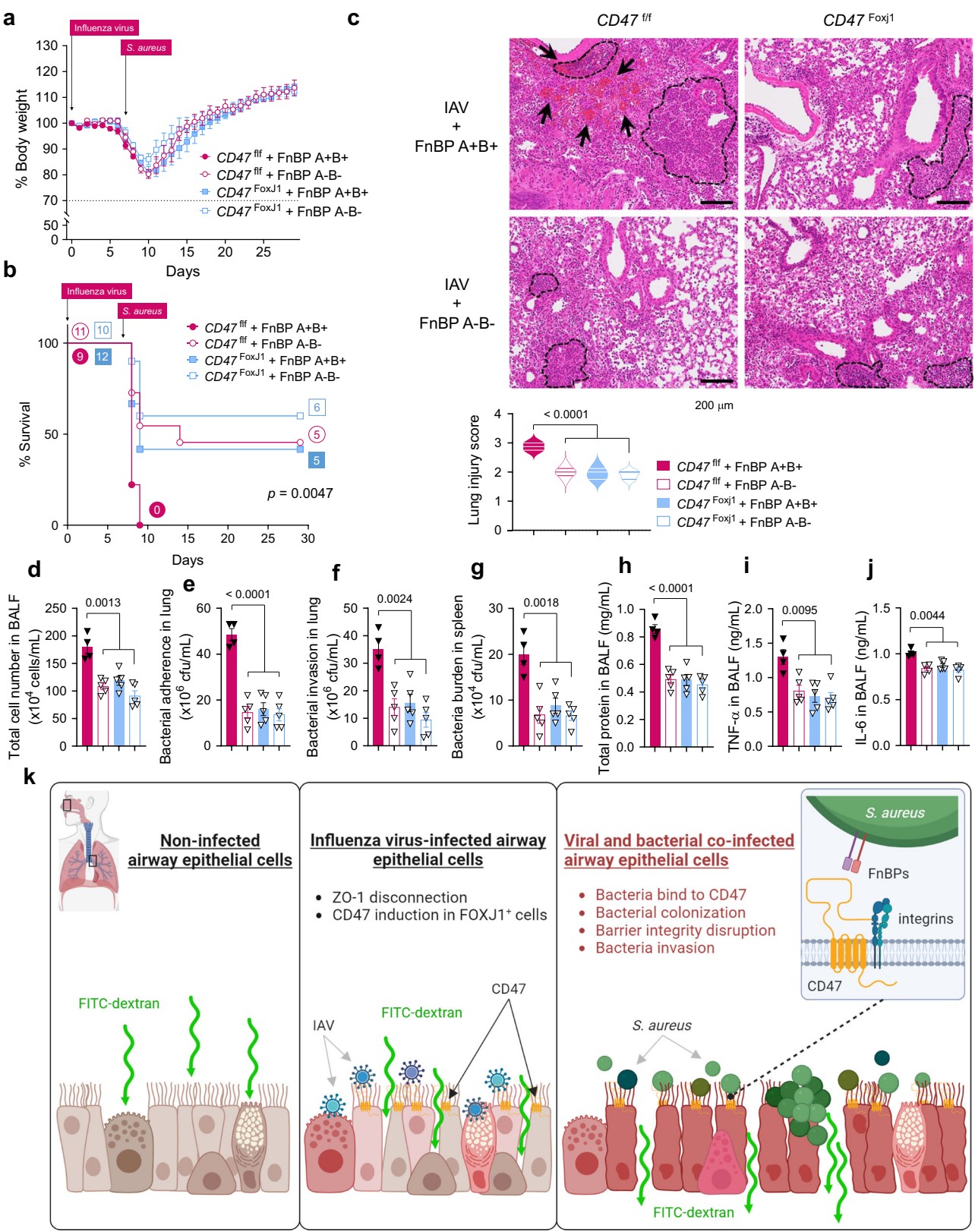

immune cell communication during bacterial infection, our findings emphasize a distinct aspect where CD47 expressed in epithelial cells during viral infection serves as an attachment site for secondary bacterial infection. Hence, it is conceivable that the roles of CD47 on epithelial cells versus myeloid cells are distinct, necessitating further investigation to fully understand these differences.

In summary, we have established that influenza virus triggers the expression of CD47 on the surface of ciliated cells in an NF-κB-dependent manner, creating an environment that promotes the attachment and proliferation of opportunistic pathogens. Furthermore, our research has shown that inhibiting the upregulated airway epithelial CD47 during viral infection can mitigate the detrimental

**Fig. 6 | Interaction between epithelial CD47 and bacterial FnBP is essential to cause super-infection. a**–**j** 6–8-weeks-old (18–21 g of body weight) *FoxJ1*-Cre; floxed (*Cd47*^Foxjl) and control floxed (*Cd47*^f/f) mice were infected with 100 PFU of influenza virus on day 0, and 1 ×10⁸ CFU of *S. aureus* WT (FnBP A+/B+) and double deletion mutant (FnBP A−/B−) on day 7 (*Cd47*^f/f + FnBP A+/B+, *n* = 9; *Cd47*^f/f + FnBP A−/B−, *n* = 11; *Cd47*^Foxjl + FnBP A+/B+, *n* = 12; *Cd47*^Foxjl +FnBP A−/B−, *n* = 10). Body weight loss (**a**) and survival rates (**b**) were monitored for 29 days. The dotted line indicates the body weight exclusion cut-off. A mantel cox survival analysis was used to compare the survival rates between groups. The numbers within circles or squares represent the count of surviving mice. Representative hematoxylin and eosin (H&E) staining of lung sections (**c**). Lung injury scores are presented as violin plots (*Cd47*^f/f + FnBP A+/B+, *n* = 4; *Cd47*^f/f + FnBP A−/B−, *n* = 5; *Cd47*^Foxjl + FnBP A+/B+, *n* = 5; *Cd47*^Foxjl +FnBP A−/B−, *n* = 5). Tissue injury parameters were measured at 24 h after bacterial infection (*Cd47*^f/f + FnBP A+/B+, *n* = 4; *Cd47*^f/f + FnBP A−/B−, *n* = 5; *Cd47*^Foxjl + FnBP A+/B+, *n* = 5; *Cd47*^Foxjl +FnBP A−/B−, *n* = 5); total cell number in BAL fluids (BALF) (**d**), bacterial adherence (**e**) and invasion (**f**) in the lung, and bacterial burden in the spleen (**g**), total protein concentrations in BALF (**h**), and inflammatory cytokines TNF-α (**i**) and IL-6 (**j**) in BALF. **k** Proposed model of viral infection-induced CD47-mediated secondary bacterial infection. Data are presented as mean values ± SEM. Significance was determined by one-way ANOVA with Tukey's multiple comparisons test. Source data are provided as a Source Data file.

consequences of subsequent bacterial infection. These findings offer fresh perspectives on the susceptibility of virus-infected individuals to secondary bacterial infections and propose CD47 as an innovative and promising therapeutic target for super-infection.

## Methods

### Ethics statement

All experiments using human nasal epithelial cells (HNECs) were approved by the institutional review board of Yonsei University College of Medicine (4-2016-1153 and 4-2021-0573), and all participants provided informed consent. All animal work was approved by the Institutional Animal Care and Use Committee (IACUC) at Yonsei University College of Medicine (protocol number 2022-0257), according to guidelines outlined by the Association for Assessment and Accreditation of Laboratory Animal Care (AAALAC) International (facility number 001071).

### Cell culture

HNECs were cultured as previously described[64], while human bronchial epithelial cells (HBECs) were cultured according to the manufacturer's instructions (CC-2540S; Lonza). HBECs used in this study were obtained from four non-asthmatic adult donors: a 43-year-old male (Hispanic), a 52-year-old male (Hispanic), a 66-year-old male (Hispanic), and a 48-year-old female (Caucasian). Briefly, the cells were seeded and expanded in T-75 flasks using bronchial epithelial growth media (BEGM, CC-3170; Lonza) containing the recommended supplements BEGM™ SingleQuots™ Kit (CC4175; Lonza) at 37 °C and 5% CO₂ in a humidified incubator. After 80% confluence was reached, the cells were detached using ReagentPack™ (CC-5034; Lonza) and seeded at a density of $5 \times 10^4$ cells on rat tail collagen (354236; BD Biosciences)-coated Transwell inserts (6.5 mm, 0.4 µm pore; 3470; Corning). The cells were cultured using BEGM in both apical and basolateral compartments until the cells became confluent (approximately 3 days). Once confluent, the cells were placed under air–liquid interface (ALI) culture conditions; the apical compartments were exposed to air, and the basolateral compartments were supported by bronchial ALI (B-ALI) differentiation media containing the recommended supplements BEGM™ SingleQuots™ Kit. The B-ALI differentiation media were changed on alternative days for 21 days. Madin-Darby canine kidney (MDCK) cells were obtained from American Type Culture Collection (CCL-34, ATCC). MDCK cells were incubated in 37 °C and 5% CO₂ in a humidified incubator using Eagle's minimum essential medium (30-2003; ATCC) supplemented with 10% fetal bovine serum (FBS) (16000-044; Thermo Scientific) and 1% penicillin–streptomycin (15140-122; Thermo Scientific). The passage number for all MDCK cells used in this study was less than 50. A549 cells were obtained from ATCC (CCL-185) and grown at 37 °C and 5% CO₂ in a humidified incubator using Dulbecco's modified Eagle medium (DMEM) (BE12-604F; Lonza) supplemented with 10% FBS and 1% penicillin–streptomycin. For all influenza virus infections, the medium used in this study was DMEM supplemented with 5% FBS and 1% penicillin–streptomycin. For super-infection, DMEM supplemented with 5% FBS without antibiotics was used. In the *NF-κB* pathway inhibition experiment, HBECs were treated with dimethyl sulfoxide (DMSO) or 10 µM of caffeic acid phenethyl ester (c8221, Sigma Aldrich) for 1 h before viral infection.

### Virus and bacteria

Influenza A virus /Korea/01/2009 (pH1N1) used in this study was propagated in embryonated chicken eggs[65]. Viruses were harvested via centrifugation of allantoic fluid or culture medium at $1300 \times g$ for 10 min. They were stored at −80 °C, and viral titers were determined in a plaque assay[65]. Briefly, MDCK cells in a monolayer were inoculated with serially diluted viral stocks for 1 h. After adsorption for 1 h, the unbound viral particles were removed, and the cells were overlaid with Minimum Essential Medium (MEM) containing 1% agarose and 1 µg/mL of tosyl phenylalanyl chloromethyl ketone (TPCK)-treated trypsin (Sigma-Aldrich). After incubation for 72 h, the cells were stained with 1% crystal violet, and viral titers were determined by counting the plaques. *S. aureus* (ATCC® 29213™) and *S. pneumoniae* (NCCP 14774) were cultured on a Bacto™ brain heart infusion (BHI; BD Biosciences) agar plate at 37 °C[66]. For the culture of GFP-tagged *S. aureus* strain, 2.5 mg/mL of tetracycline-hydrochloride (Sigma-Aldrich) was added. *S. aureus* laboratory strain 8325-4 (FnBP A+/B+) and three mutant strains (FnBP A+/B−, A−/B+, and A−/B−) were grown as previously described[47]. *P. aeruginosa* and *S. epidermidis* were cultured on Luria-Bertani (LB) agar plates and Tryptic Soy Agar (TSA) plates, respectively, at 37 °C.

For in vitro pull-down assay, we utilized five *S. aureus* mutant strains of cell wall-anchored proteins (CWAs), namely JE2 *fnbB::Tn* (NE728), JE2 *sasG::Tn* (NE825), JE2 *isdB::Tn* (NE1102), JE2 *sdrE::Tn* (NE98), and JE2 *clfA::Tn* (NE543), all of which were obtained from the Nebraska Transposon Mutant Library (BEI resources)[67]. The absence of CWAs in these mutants was validated through PCR using specific primers. The wild-type (WT) control used for this study was the Methicillin-resistant *S. aureus* (MRSA) strain USA300 JE2. The WT JE2 strain was cultured at 37 °C on TSA plates, while the other mutant strains were cultured at 37 °C on TSA plates supplemented with 10 µg/mL erythromycin.

### Mice

C57BL/6J background *CD47* floxed (*Cd47*^f/f) mice were purchased from the Mouse Biology Program at UC Davis, CA, USA (*Cd47*^tm1c(KOMP)Mbp/Mmucd, 046999-UCD). *Rosa26-tdTomato* reporter mice were kindly provided by Dr. Jinwoong Bok (Yonsei University College of Medicine, Seoul, Republic of Korea). *FoxJ1*^Cre transgenic and *LysM*^Cre knock-in/knock-out mice were kindly provided by Dr. Michael J. Holtzman (Washington University School of Medicine, St. Louis, MO) and Dr. Hyoung-Pyo Kim (Yonsei University College of Medicine, Seoul, Republic of Korea), respectively. *FoxJ1*^Cre;*Cd47*^f/f (*Cd47*^Foxjl), *LysM*^Cre;*Cd47*^f/f (*Cd47*^LysM), and *Cd47*^f/f mice were bred in-house at Yonsei University College of Medicine in Seoul, Republic of Korea. Mice were housed in the specific pathogen-free (SPF) animal facilities under a 12 h light-dark cycle at 20 ± 2 °C, a humidity of 50 ± 5%, ventilation of 10–15/h, the light of 150–300 Lux, noise of less than 60 dB and maintained on normal chow diet. The genotypes of mice were determined using PCR amplification of tail DNA using primers listed in Supplementary Table 1.

## Viral and bacterial super-infection (in vitro model)

For in vitro super-infection experiments, fully differentiated HNECs or HBECs were infected with phosphate-buffered saline (PBS) or influenza virus at a multiplicity of infection (MOI) 1 and incubated for 2 h. After viral infection, cells were washed twice with PBS and supplied with fresh media. The following day, the virus-infected cells were infected with PBS or *S. aureus* at MOI 5 (HNECs) or at MOI 3 (HBECs) or *S. pneumoniae* at MOI 5 (HBECs), and incubated for 3 h. After bacterial infection, cells were washed twice with PBS and supplied with fresh media supplemented without gentamycin.

## Viral and bacterial super-infection (in vivo model)

Six to seven-week-old mice (18–21 g body weight) were used. Mice were anesthetized by intraperitoneal injection (i.p.) of Zoletil: Rompun: PBS (1:1:8) mixture (100 μL) and infected intranasally with PBS or 100 plaque-forming units (PFUs) of influenza virus (30 μL) at day 0. For the super-infection model, uninfected or virus-infected mice were infected intranasally with PBS or $1 \times 10^8$ CFU of *S. aureus* (20 μL) at day 7. Following viral infection, mice were monitored daily for changes in weight and mortality until day 29. Mice manifesting a weight loss exceeding 30% of their initial body weight were humanely euthanized and subsequently excluded from further analysis in both weight loss and survival data, in accordance with the humane euthanasia criteria defined by the relevant animal protocols. Weight loss (or discovery of mice found dead) was the sole criterion for determining mortality in our study. Clinical scores were not used as an additional parameter for declaring a mouse dead.

## Proteomic analysis

iTRAQ or tandem mass tag (TMT) proteomic methods were used to identify total protein expression differences between uninfected or influenza virus-infected HNECs. Cells were lysed in RIPA buffer (Sigma Aldrich) with Halt™ Protease & Phosphatase Inhibitor Single-Use Cocktail, EDTA-free (100x) mixture (78443; Thermo Scientific). Acetone-precipitated protein pellets were stored at −80 °C. Proteins from HNECs or HBECs were labeled using the 8plex iTRAQ labeling kit (4390812; SCIEX) or the TMT 10plex isobaric label reagent set (90111; Thermo Scientific), respectively, according to the manufacturer's instructions. The labeled peptides were pooled and then dried using SpeedVac (Thermo Scientific). Strong cation exchange fractionation, liquid chromatography (LC)-mass spectrometry (MS), database searching, and quantitative data analysis were performed by Poochon Scientific[68].

## Real-time quantitative PCR (RT-qPCR)

Total RNA was isolated from HNECs, HBECs, and mouse lung tissues using Hybrid-R™ (305-101; GeneAll Biotechnology). cDNA was synthesized via reverse-transcription using 500 ng of RNA with random hexamer primers (N8080127; Invitrogen), RNase inhibitor (N8080119; Applied Biosystems), dNTPs (N8080260; Applied Biosystems), and M-MLV reverse transcriptase (28025013; Invitrogen). For real-time qPCR, KAPA SYBR FAST qPCR master mix (2X) was used according to the manufacturer's instruction. PCR reaction was performed using QuantStudio 3 Real-Time PCR System (Thermo Scientific). The gene expression levels were evaluated using the comparative Ct method ($2^{-\Delta\Delta Ct}$ method). Primers used for real-time qPCR in this study are listed in Supplementary Table 1.

## Confocal microscopy

For immunofluorescence (IF) staining, cell slides were fixed with 4% paraformaldehyde (PFA) for 20 min, permeabilized with 0.1% Triton X-100 (0694; Amresco, NY, USA) in PBS for 5 min, and treated with 50 mM $NH_4Cl$ (A9434; Sigma Aldrich) in PBS for 15 min. After incubation in a blocking solution containing 0.1% bovine serum albumin (BSA), 0.1% Triton X-100, 1% normal donkey serum (NDS) for 1 h at room temperature (RT), the slides were incubated with primary antibodies diluted in Dako antibody diluent (S3022; Dako) at 4 °C overnight and with secondary antibodies in diluted in Dako antibody diluent at RT for 1 h. For mouse trachea IF staining, trachea was fixed in 4% PFA for 24 h. Trachea samples were transferred to 50% OCT compound in PBS for 24 h, sequentially transferred to 100% OCT compound for 24 h, embedded in Tissue-Tek OCT compound (4583; Sakura Finetek), snap-frozen in liquid nitrogen, and stored at −80 °C. All sections were cut at 8 μm in a cryostat microtome. Cryo-sections on slides were covered with 3% peroxidase blocking solution (S2023; Dako) for 10 min at RT. After the inactivation of endogenous peroxidase, slides were washed twice with 1% TBS. Blocking was conducted using 5% BSA at RT for 1 h. The slides were incubated with primary antibodies (1:100) diluted in Dako antibody diluent at 4 °C overnight. The following day, slides were washed three times with 1% TBS and then incubated with secondary antibodies diluted with Dako antibody diluent at RT for 1 h. After final washing, the slides were cover-slipped with Fluoroshield™ and DAPI (F6057; Sigma Aldrich). Confocal images were acquired by a confocal laser-scanning microscopy (LSM980; Carl Zeiss Microscopy GmbH) and analyzed by ZEN image software (Carl Zeiss, ZEN 3.0 lite). Cytospin experiments were conducted as previously described[64]. Primary antibodies used in IF staining as follows: rabbit anti-ZO-1 (61-7300; Thermo Scientific), mouse anti-hCD47 (MA5-11895; Thermo Scientific), goat anti-mCD47 (AF1866; R&D Systems), rabbit anti-α-tubulin (Acetyl K40, ab179484; Abcam), rabbit anti-MUC5AC (61193S, Cell Signaling Technology), rabbit polyclonal anti-Influenza A NP (PA5-32242, Thermo Scientific), mouse monoclonal anti-p63 (sc-25268, Santa Cruz Biotechnology), and rabbit anti-*S. aureus* (ab20920; Abcam). Secondary antibodies used in IF staining as follows: Alexa Fluor 488 donkey anti-mouse IgG (A21202; Molecular Probes), Alexa Fluor 488 donkey anti-rabbit IgG (A21206; Molecular Probes), Alexa Fluor 568 goat anti-mouse IgG (A11004; Molecular Probes), Alexa Fluor 568 goat anti-rabbit IgG (A11011; Molecular Probes), and Alexa Fluor 647 donkey anti-goat IgG (A21447; Molecular Probes).

## Flow cytometry

Cells were detached from culture plates and separated using a mixture of non-enzymatic cell dissociation solution (C5914; Sigma Aldrich) and trypsin-EDTA in a 1:1 ratio. Cells were incubated with 1 μg of mouse IgG isotype control antibody (31903; Invitrogen) or mouse anti-hCD47 APC-conjugated antibody (ab134485; Abcam, 1:200) for 20 min on ice. Cells were analyzed using the BD FACSLyric™ (BD Biosciences). Non-viable cells were excluded from further analyses. Data were acquired with BD FACSuite v1.3 (BD Biosciences) and analyzed with FlowJo™ v10.6.2 software (BD Biosciences).

## Immunoblotting

Primary epithelial cells and mouse lung tissues were lysed in RIPA buffer with Halt™ Protease & Phosphatase Inhibitor Single-Use Cocktail, EDTA-free (100x). Equal concentrations of proteins (25 μg) were separated using 8–12% SDS-PAGE gels and transferred to PVDF membranes (Merck Millipore, Burlington, MA, USA). After blocking with 5% skim-milk or bovine serum albumin (BSA) in Tris-buffered saline (50 mM Tris-Cl, pH 7.5, 150 mM NaCl) containing 0.5% Tween 20 (TTBS) for 1 h at RT, the membranes were incubated with primary antibodies (1:1,000) in 5% skim-milk or BSA with TTBS at 4 °C overnight. The membranes were then washed three times with TTBS and incubated with secondary antibodies diluted (1:1,000) in 5% skim-milk with TTBS for 1 h. Blots were visualized using Pierce ECL Western Blotting Substrate (Thermo Scientific) and exposed to X-ray film. Primary antibodies used in immunoblotting as follows: rabbit anti-ZO-1 antibody (61-7300; Thermo Scientific, 1:500), mouse anti-ZO-1 (33-9100; Thermo Scientific, 1:1,000), mouse anti-CD47 antibody (MA5-

11895; Thermo Scientific, 1:500), mouse anti-ICAM-1 (sc8439; Santa Cruz Biotechnology, 1:1,000), rabbit anti-p65 (4764S; Cell Signaling Technology, 1:1,000), rabbit anti-phospho-p65 (Ser536) (3033S; Cell Signaling Technology, 1:1,000), mouse monoclonal anti-CD47 (AF1866; R&D Systems, 1:500), mouse anti-β-actin (sc-47778; Santa Cruz Biotechnology, 1:2,000) and mouse anti-GAPDH (sc-32233; Santa Cruz Biotechnology, 1:2,000). Secondary antibodies used as follows: Peroxidase AffiniPure goat anti-rabbit IgG (H + L) (111-035-003; Jackson ImmunoResearch, 1:1,000), mouse anti-goat IgG-HRP (sc-2354; Santa Cruz Biotechnology, 1:500), and goat anti-mouse IgG (H + L)-HRP (SA001; GenDEPOT, 1:1,000).

### Paracellular permeability
After infection or treatment, transwell inserts were transferred to new cell culture plates (Corning). After two PBS washes, the apical compartments of the transwell were treated with 4 kDa FITC-dextran (Sigma Aldrich), and the basolateral compartments of the transwell were replaced with 1 mL of Hank's balanced salt solution (HBSS, LB003-02; Welgene). After incubation for 2 h, the HBSS of the basolateral compartments was collected and analyzed using BioTek Cytation™ 5 Cell Imaging Multi-Mode Reader (Agilent).

### Measurement of transepithelial electrical resistance (TEER)
The electric resistance of the respiratory epithelium was determined by measuring TEER in HNECs or HBECs using the EVOM® resistance meter and Endohm® chamber (World Precision Instruments, Sarasota, FL, USA) as described previously[69]. The values for cell-covered transwell inserts were expressed in standard units of ohms per square centimeter ($\Omega/cm^2$) after subtracting the resistance of blank transwell inserts and presented as mean ± standard error of mean (SEM).

### Lactate Dehydrogenase (LDH) assay
Following infection, cell cytotoxicity was measured using CytoTox 96 Non-Radioactive Cytotoxicity Assay kit (G1780; Promega) according to the manufacturer's instructions. Briefly, apical and basolateral supernatants of HNECs and HBECs were collected and incubated with the reagent for 30 min in a 96-well plate. Results were obtained at 490 nm using VersaMax™ Microplate Reader (Molecular Devices). Positive control was supplied with the kit, while PBS or culture medium was used as the negative control. Data was analyzed with SoftMax® Pro Software v5.2 (Molecular Devices).

### Short hairpin RNA (shRNA) transfection
For the knock-down of human *CD47*, a plasmid-based lentiviral shRNA was purchased from Lugen Sci. The sequence for the *CD47*-specific shRNA was 5′-GCA TGG CCC TCT TCT GAT TTC-3′, and that for scrambled shRNA (sc-shRNA) was 5′-GCA CTA CCA GAG CTA ACT CAG ATA GTA CT-3′. HNECs or HBECs (passage #2) were transfected with lentiviral shRNAs (MOI 10) using polybrene reagent (Lugen Sci). After a 24-h incubation, cells were supplied with fresh media. Once confluent, the cells were placed under ALI culture conditions; the apical compartments were exposed to air, and the medium for basolateral compartments was changed every other day for 14 days (HNECs) or 21 days (HBECs).

### In vitro pull-down assay
His-tagged human CD47 protein (250 ng) (CD7-H5227; Acro biosystems) and $1 \times 10^2$ colony-forming units (CFUs) of bacteria were mixed, and the mixtures were isolated using Dynabeads™ His-Tag Isolation & Pulldown kit (10103D; Invitrogen) according to the manufacturer's instructions. After placing the tube on DynaMag™-2 magnet (12321D; Invitrogen), the beads were washed twice with 1 × binding/wash buffer, and the supernatant was removed. The washed buffers (*supernatants*) and the final beads (*pellets*) were plated on a BHI agar plate and TSA plate containing 10 μg/mL erythromycin, separated using 12% SDS-

PAGE gel, and blotted with anti-human CD47 antibody (B6H12.2, MA5-11895, Thermo Scientific, 1:500).

### Bacterial adhesion/invasion assay
HBECs and A549 cells were infected with PBS or influenza virus at MOI 1. After incubation for 2 h, cells were washed twice with PBS and supplied with fresh media. The following day, the virus-infected cells were subjected to PBS or infected with *S. aureus* at MOI 10 (A549) or *S. aureus* laboratory strain 8325-4 (FnBP A+/B+) and three mutant strains (FnBP A+/B−, A−/B+, and A−/B−) at MOI 3 (HBECs). After incubation for 3 h, cells were washed twice with PBS to remove unbound bacteria and lysed with 0.8 mL of 0.025% Triton X-100. After serial dilutions with PBS, the number of bacteria bound to the cells was quantified[61]. For CD47 neutralization, virus-infected cells were treated with 5 μg/mL of IgG1 control antibodies (31903; Invitrogen) or 5 μg/mL of α-hCD47 neutralizing antibodies (MA5-11895; Invitrogen) for 2 h, followed by bacteria infection for 3 h. For in vivo bacterial adhesion assay, the infected mice were euthanized, and the lung was collected. The lung tissue was homogenized in PBS and centrifuged at $500 \times g$ for 5 min. Following centrifugation, the supernatant was discarded, and the sediments were re-suspended with 0.025% Triton X-100. Subsequently, after serial dilutions with PBS, the number of bacteria bound to the cells was quantified[61]. For in vivo bacterial invasion assay, the lung tissue was incubated with PBS containing 1% penicillin-streptomycin at 37 °C with 5% $CO_2$ for 1 h to remove extracellular and surface-adherent bacteria. After the incubation period, the lung tissue was homogenized and centrifuged, and then the sediments were re-suspended with 0.025% Triton X-100. Following serial dilutions with PBS, the number of bacteria bound to the cells was quantified[61].

### Measurement of airway hyper-responsiveness (AHR)
Airway hyper-responsiveness (AHR) was measured as the change in airway function using whole-body plethysmography (Buxco Electronics Ltd., USA) as previously described[70]. Briefly, mice were placed unrestrained in a chamber connecting a transducer to measure pressure changes inside the chamber. After acclimation, increasing concentrations of methacholine (0, 6.25, 12.5, 25, and 50 mg/mL) were nebulized for 3 min, and enhanced pause (Penh, a dimensionless unit, correlates with pulmonary resistance) was measured for 2 min using FinePoint software (Buxco Electronics Ltd.).

### Histologic analysis
Lung tissues were fixed in 4% PFA for 24 h and embedded in paraffin. The mouse lung tissues were sectioned and stained with hematoxylin and eosin (H&E) to count inflammatory cells that infiltrated the lungs. Lung injury was scored as previously described[71]. Briefly, blinded adults were scored using lung sections stained with H&E. The severity of inflammation and pneumonia was evaluated based on the number of perivascular lymphocytic infiltration, peribronchial lymphocytic infiltration, bronchitis, and alveolar hemorrhage using the following scoring system: 0, absence; 1, mild; 2, moderate; and 3, severe.

### Cell count, protein quantification, and cytokine ELISA
After whole-body perfusion, the mouse right lung lobe and the spleen were used for quantification of bacterial CFU, and left lobe was used for H&E staining for counting inflammatory cells. BAL fluids were collected from the lungs after washing with ice-cold PBS (1 mL). After centrifugation at $1700 \times g$ for 5 min, the supernatants were collected and stored at −80 °C. The pellets of BAL fluid were resuspended with 1 mL of 2% FBS in PBS and then counted using EVE™ automated cell counter (NanoEnTek). Total protein concentration in mouse BAL fluid was measured using Pierce™ BCA protein assay kit according to the manufacturer's instructions (23225; Thermo Scientific). Levels of IL-6

(DY406) and TNF-α (DY410) in mouse BAL fluid were quantified using DuoSet® ELISA kit (R&D Systems, Minneapolis, MN, USA), according to the manufacturer's instructions. The ELISA detection limits for IL-6 and TNF-α were consistently greater than pg/mL. The sample absorbance was measured at 450 nm using a VersaMax™ Microplate Reader (Molecular Devices). Data was analyzed with SoftMax® Pro Software v5.2 (Molecular Devices).

### In vivo neutralizing assay
C57BL/6 mice were purchased from Orientbio. Six to eight-week-old male mice (18–21 g body weight) were used. Mice were anesthetized and infected intranasally with 10 PFU of influenza virus at day 0. On days 5 and 7 after viral infection, mice were injected intranasally with either IgG2a control antibodies (clone 2A3; Cat.BE0089; BioXCell) or anti-mouse CD47 antibodies (clone MIAP301; Cat.BE0270; BioXCell). On day 7, 30 min after the second treatment of antibodies, mice were infected intranasally with $5 \times 10^5$ CFU of $S. aureus$.

### Interferon stimulation assay
For type I and III interferon (IFN) stimulations, basolateral region of hBECs were treated with recombinant human (rh) IFN-β protein (8499-IF; R&D Systems) using 0.1, 1, 10, 100, 1000 U/mL and recombinant human (rh) IL-29/IFN-λ1 protein (1598-IL; R&D Systems) using 0.1, 1, 10, 100 ng/mL for 24 h, respectively. To validate IFN stimulation, the induction of interferon-stimulated genes (ISGs) were analyzed by qPCR. To block type I IFN signaling experiment, 2 μg/mL of IFNAR1 (ab10739; abcam, 1:50) and normal goat IgG control antibody (AB-108-C; R&D Systems, 1:500) were treated with basolateral region for 1 h prior to the rhIFN-β protein treatment. Similarly, to block type III IFN signaling experiment, 10 μg/mL of IFNλR1 (PBL assay science; 21885-1, 1:50), mouse IgG isotype control antibody (31903; Invitrogen, 1:500) were treated with basolateral region for 1 h prior to the rhIFN-λ1 protein treatment.

### Quantification and statistical analysis
Data are analyzed by GraphPad Prism v10.1.0 software and expressed as mean ± SEM. A Student's $t$ test was employed for single-variable comparisons, one-way ANOVA was utilized for multiple comparisons, and two-way ANOVA was applied for analyses involving multiple variables. Values of $p$ less than 0.05 were considered statistically significant.

### Reporting summary
Further information on research design is available in the Nature Portfolio Reporting Summary linked to this article.

## Data availability
Data presented in the manuscript will be made available to investigators following request. The raw data files of the iTRAQ dataset are available in the ProteomeXchange Consortium via the PRIDE partner repository under accession code PXD042618 (http://proteomecentral. proteomexchange.org). Proteomics data analysis was performed using the Database for Annotation, Visualization and Integrated Discovery (DAVID) v6.8 (https://david.ncifcrf.gov/tools.jsp). All relevant data are included in the manuscript and the Supplementary Information. Source data are provided with this paper.

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

## Acknowledgements

We thank T. Foster (Trinity College Dublin, Ireland) for providing *S. aureus* strains; J. Bok (Yonsei University College of Medicine, Korea) for providing Rosa26-tdTomato reporter mice; M.J. Holtzman (Washington University School of Medicine, St. Louis, MO) for providing *Foxj1* Cre transgenic mice; and H.-P. Kim (Yonsei University College of Medicine, Korea) for providing *LysM* Cre knock-in/knock-out mice. This work was supported by Bio & Medical Technology Development Program (2019M3A9B6066971 to J.-H.R.; 2021M3H9A1030260 to J.K.S.), Basic Science Research Program (RS-2023-00250400 to Y.W.C.), National Creative Research Initiative programs (2015R1A3A2033475 to W.J.L.) and R&D Project for the Korea Mouse Phenotyping Center (2013M3A9D5072550 to J.K.S.; 2016M3A9D5A01952415 to J.-H.R.) of the National Research Foundation of Korea (NRF), which is funded by the Ministry of Science and ICT. Graphical illustration in Fig. 6k was created with BioRender.com under academic subscription.

## Author contributions

S.M., J.-H.Y., Y.W.C., and J.-H.R. conceived the project; S.M. conducted most of the experiments; S.H., J.R and Y.W.C. helped with experiments and data analysis; M.-S.R., H.-J.C., and C.-H.K. provided clinical samples; I.-H.J., S.S.Y., K.T.N., M.-S.P., J.K.S., W.-J.L., and Y.W.C. provided critical reagents; S.M., Y.W.C., and J.-H.R. wrote the manuscript; Y.W.C., and J.-H.R. supervised the study.

## Competing interests

The authors declare no competing interests.
