## [Peer Review File · Nature Communications]

Airway epithelial CD47 plays a critical role in inducing influenza virus-mediated bacterial super-infectionEditorial Note: Parts of this Peer Review File have been redacted as indicated to remove third-party material where no permission to publish could be obtained.

REVIEWER COMMENTS

Reviewer #1 (Remarks to the Author):

The authors show here that influenza virus infection of nasal or bronchial epithelia leads to upregulation of CD47 which serves facilitates bacterial adhesion and epithelial barrier disruption. In a murine superinfection model, the authors claim that this leads to increased mortality which can be blocked genetically or by blocking Abs targeting CD47.

The concepts of CD47 induction by viruses, the increase of bacterial adhesion factors by viral infection, and the facilitation of Staph. aureus infection are not new, thus limiting the novelty of this study. However, the authors go at great lengths to show that CD47-mediated facilitated bacterial adhesion explains the bacterial superinfection phenotype. Major concerns preclude publication as it stands:

1. The mouse coinfection model: As shown in fig. S9 a S.a. dose is used that practically kills the mice on their own. Why such a high dose? Most coinfection models show strong synergy, so there is no need to have such similarity of morbidity in single bacterial and superinfection. One consequence is that very small weight loss differences between single bacterial and double infection translate into major survival differences, is this not a bit artefactual depending on how death is defined? When do you declare a mouse as dead?

This also means that all treatments / targeting strategies do not look very different in terms of weight loss, but appear to generate an exaggerated difference when looking at mortality. This applies to fig. 4j, that shows no effect, so the apparent significance in survival in 4k must come from absolutely minor cutoff differences of the weight loss – or clinical scores? Therefore, overall the survival differences upon treatment remain unconvincing.

2. What leads from bacterial adhesion to barrier loss? The authors show that ZO-1 protein levels and distribution are perturbed by virus infection, for many researchers this alone would be sufficient (and maybe superior) to explain increased bacterial burden – as this defines proper barrier function. In contrast to this, the mechanism described here should lead to accumulation of bacteria without explaining how they actually breach the barrier more efficiently. However, in the mouse experiments, bacterial burden is unaffected upon treatment (fig. 5e, n). Even if the authors added 10 more mice per group and reached statistical significance, the less than 2 fold difference in bacterial load is much less than what described in other models of coinfection where often several logs difference are found. Therefore, the whole narrative is not quite convincing for me.

3. There is a woeful lack of quantification in this paper when histology and WB data is shown: fig.s 1a, 1b, 1e, 2b, 3c, S8, to name just the first that spring to mind. Without more serious quantification using biological replicates, the data shown here is too anecdotal.

4. Fig. S1, Proteomics: ALI cultures are mixed cultures, how do the authors know that protein changes are not due to changes in cell representation?

5. Fig. S2: How can the FC of CD47 be 1.15 (as stated in the text) and yet the WB and microscopy shows huge differences, certainly more than a 1.15 increase?

6. The NFkB blockade could block CD47 indirectly: IFN induction is NFkB-dependent, is CD47 induction IFN dependent? Use IFNAR/IFNLR blocking Abs.

7. Not all cells within the epithelial cultures express CD47, are those who do virus-infected? Double staining for viral protein and CD47.

8. Text reference to fig. 2e, accepted m/s: does this mean that the data is not yet publicly available?

9. Fig. S4: In our own experience, when cultures really are as they look in S4d and h, then the TEER is zero or near zero. Here, huge holes appear to be visible in the culture which would lead to a complete collapse of the resistance. Also, are these cultures done without antibiotics? The bacteria must take over completely by d5 and even more so by d7 which would perturb the tissue

culture system to a point where cell death will occur that is not representative of physiological processes. In this figure, the p values of double infected vs single infected are important as the argument is that there is a synergy, so comparison to mock alone is insufficient.

10. How do you explain the discrepancy between 2a, b and c? There is an only twofold reduction in mRNA which however leads to disappearance of protein in the WB (but not quantification, n=1?), and a massive increase in permeability. Similar, S5A, C?

11. Throughout the study, both binding via CD47 and the involvement of fibronectin binding protein is shown only indirectly. At no point is direct interaction between bacteria and CD47 or pull downs of CD47 by the bacteria shown. Similarly, using FnBP mutants to show the dependency on fn is an elegant but inconclusive way to show involvement of fn or indeed the binding protein – these mutants may have additional defects, leading indirectly to the phenotypes shown here. Along these lines, fig. 3h shows a laudable effect that however needs another orthogonal control, for instance another, unrelated protein.

Minor concerns:

1. Mx1 and Mx2 are not supposed to be membrane-bound... nuclear or cytosolic
2. line 105 CEACAM typo
3. line 135 where do you show us that this is on the apical side?
4. Fig. S4G, so most cells are dead? How do you translate LDH levels into % viability?
5. Figure S6, the authors need to show single infection with control shRNA.
6. line 189 and later, e.g. 211: The authors need to explain in the main text how bacterial adhesion is measured.
7. In line 189 and 198 the authors cannot say here that this is adherence to CD47 as they did not test this here. It is adherence to the cells, and it is CD47 dependent which may be direct or indirect at this point.
8. Fig. S10a,b, what is shown here? The M and f nomenclature is not in text or legend. Why is the CD47 band identical in b in the presence or absence of Cre?
9. Fig. S10d these virus measurements are a bit late, what about d2,3,4 p.i.?

Reviewer #2 (Remarks to the Author):

Summary – The manuscript by Moon and colleagues identifies a novel role for influenza-induced epithelial CD47 in promoting attachment of *S. aureus* in vitro and in mediating severe super-infection disease in vivo. The authors show that influenza induces CD47 in human nasal and bronchial ciliated epithelial cells via NF- κ B induction. CD47 is then shown to be required for in vitro super-infection induced epithelial barrier leak and loss of TEER. The authors then demonstrate that *S. aureus* FnBP plays a contributing role to *S. aureus* adherence via interaction with CD47. Finally, in the mouse model, epithelial specific deletion of CD47 results in decreased super-infection disease phenotype and a modest reduction in mortality. CD47 monoclonal antibody treatment mimicked these effects. Overall, this is a well designed and clearly presented study with compelling results. The manuscript could be improved as detailed below.

Major Comments –

1) Influenza-induced CD47 time course – It would be helpful if the authors could show a time course for CD47 induction during influenza in vivo. Does CD47 expression track with the window of enhanced susceptibility to secondary infection (day 5-10) post infection? This would help delineate the specific contribution of CD47 to susceptibility.

2) *S. aureus* mutants – It would be great to close the pathway loop by infecting WT and Foxj1-Cre CD47^{fl/fl} mice with the *S. aureus* lacking FnBA/B to determine if the phenotype matches to the expected model. For example in WT the mutant *S. aureus* should have decreased pathogenicity, while in the epithelial CD47 KO there should be no effect of deletion of FnBA/B. It would be helpful

to also cite Groud et al. Microbiol Spectrum 2022, which tested FnbB deletion in a similar super-infection model.

Minor Comments –

1) Line 308 – Clara Cells have been renamed to Club Cells.

Reviewer #3 (Remarks to the Author):

To the authors

The authors have performed an extensive analysis of a potential mechanism involved in *S. aureus* superinfection following flu, highlighted with the use of both FoxJ1 and CD47 LysM Cre floxed mice. While the in vitro data appear straightforward, suggesting the upregulation of CD47 in epithelial cells following influenza infection and recognition of this by *S. aureus* FNBP, the mouse models do not clearly support the underlying hypothesis. The expected WT mouse control is only provided in a supplemental figure - but not with the mouse models using Ab or floxed mice. Additional controls are necessary to adequately interpret the data shown.

1- WT controls are needed in the mouse studies. In Fig 4 - There appears to be a substantial survival effect (most mice are alive 4-5 days post *S. aureus* infection) with the administration of either the control Ab or that targeting CD47 - if one compares survival at 24 hours post Staph infection- which is a typical outcome point. This is very different than the control data showing that virtually all of the mice are dead within 24 hours following the administration of *S. aureus*. As survival is the outcome in the Supp. Fig 9.

WT mouse infection experiments using PBS as well as antibody (which non-specifically binds *S. aureus* protein A) are needed in parallel.

In Fig 4B - how many mice survived beyond day 10? It appears that the conclusions that the FoxJ1 floxed mice have increased survival may be based on a very small number of survivors, but this is impossible to tell from the data presented. Of note, the Suppl fig11 data indicates that CD47 is critical in myeloid cells - as these mice all died apparently within 24 hours post *S. aureus* infection - but this is impossible to judge without a WT mouse control is necessary. Again, mouse numbers are not provided for the time points post infection. While changes in mouse weight are outcomes for viral infection, cfus in addition to death, are typically included in bacterial models.

Previous studies (Nauseef 2014 JI) demonstrated that CD47 functions as an "eat me" signal that is critical for macrophage clearance of neutrophils that have ingested *S. aureus*. In the absence of CD47- there is increased infection. Thus, the relative roles of CD47 on epithelial versus myeloid cells may be very distinct and needs to be better addressed.

2- Characterization of the cell lines - A major point of the manuscript is the expression of CD47 by ciliated epithelial cells - thus the use of polarized, presumably ciliated cell lines is expected. The authors provide an extensive analysis of nasal epithelial cells, a site of colonization (.i.e.no immune response?) but not infection. In the suppl fig 7 CD47 appears to be predominantly expressed in "deuterosomal" cells -but is co-expressed with FoxJi to a lesser extent in multi-ciliated cells. As the nasal epithelium is not a site of *S. aureus* infection - is there a similar analysis of the human bronchial epithelial cells? What was their expression of cilia? Of CD47 and FoxJ1? These are important - since A549 cells, as a non-polarized carcinoma cell line are not especially relevant to the airway.

Why are viable *S. aureus* necessary to induce CD47? Do they require uptake and internalization? This seems distinct from a simple ligand -receptor interaction.

3- Several experiments demonstrate the loss of the epithelial barrier function in vitro. As an in vivo correlate - what were the rates of *S. aureus* bacteremia as evidence of invasion? Did the lack of CD47 protect from in vivo invasion? Was this the case with the myeloid specific KOS as well as the FoxJI floxed mice??

Reviewer #4 (Remarks to the Author):

The study is somewhat innovative in clarifying the crucial role that CD47 plays in viral-bacterial super-infection. And it has certain clinical application value, which provides new ideas for the treatment of super-infection. This manuscript has novelty and innovation. However, there were several main concerns that should be addressed as follows:

Major issues:

- 1) Introduction part of the paper is poorly cited and does not clarify why CD47 was chosen for the study. The summary of this study at the end of the introduction is not sufficiently clear.
- 2) Why is iTRAQ done 5 days after IAV infection. Please clarify the basis of timepoint selection.
- 3) Can *S. aureus* infection alone or super-infection induce CD47 expression? This is of major importance.
- 4) Compared with wild mice, there was no difference in the viral load in the lungs of IAV infected with ciliated cell-targeted CD47-deficient mice for 7 days, but after 7 days of infection, did CD47 affect the innate immunity of mice, leading to changes in viral load, and affect the severity of lung lesions?
- 5) Fig. 3: cells were washed twice with PBS to detect bacteria. The treatment would result that the detected bacteria which included not only bacteria adhered to the cells but also internalized bacteria into cells.
- 6) Line 137-138: It would be better to modify this small title to "Epithelial CD47 in HBECs and HNECs facilitates to the super-infection", etc., This would be in line with the Line 172-174. Knock-out and neutralization tests are only further validation.
- 7) Line 186-187 and Fig.3: Why was A549 cells used to detect bacterial adhesion? All the previous studies used HNEC and HBEC, and did not study the expression of CD47 in A549. Different cell lines may have an effect on the results, the author should verify the universality of the results.
- 8) Supplementary Fig. 9: Did the authors measure the viral titers except for the bacteria burden?
- 9) Line 304-315: This paper mainly focused on influenza virus, so it is not necessary to include the experimental results of SARS-CoV-2. The description of SARS-CoV-2 has been added in the first part of discussion, so there is no need to use another paragraph to elaborate.

Minor issues:

- 1) Supplementary Fig. 9a: Significance analysis is recommended.
- 2) Supplementary Fig. 9c, Supplementary Fig. 11c, Fig. 4c: The lesion location was not shown in figures.
- 3) Line 260: "Figure. S10A" should be modified to "Supplementary Fig. 10a".

Reviewer #1 (Remarks to the Author):

The authors show here that influenza virus infection of nasal or bronchial epithelia leads to upregulation of CD47 which serves facilitates bacterial adhesion and epithelial barrier disruption. In a murine superinfection model, the authors claim that this leads to increased mortality which can be blocked genetically or by blocking Abs targeting CD47.

The concepts of CD47 induction by viruses, the increase of bacterial adhesion factors by viral infection, and the facilitation of *Staph. aureus* infection are not new, thus limiting the novelty of this study. However, the authors go at great lengths to show that CD47-mediated facilitated bacterial adhesion explains the bacterial superinfection phenotype. Major concerns preclude publication as it stands:

1. The mouse coinfection model: As shown in fig. S9 a *S.a.* dose is used that practically kills the mice on their own. Why such a high dose? Most coinfection models show strong synergy, so there is no need to have such similarity of morbidity in single bacterial and superinfection. One consequence is that very small weight loss differences between single bacterial and double infection translate into major survival differences, is this not a bit artefactual depending on how death is defined? When do you declare a mouse as dead?

This also means that all treatments / targeting strategies do not look very different in terms of weight loss, but appear to generate an exaggerated difference when looking at mortality. This applies to fig. 4j, that shows no effect, so the apparent significance in survival in 4k must come from absolutely minor cutoff differences of the weight loss – or clinical scores? Therefore, overall the survival differences upon treatment remain unconvincing.

[Response 1-1] First and foremost, we apologize for the lack of clarity in our description of the experimental setup for our mouse super-infection model. Its lack of clarity resulted in numerous questions from the reviewers (Please be aware that all figure numbers in the responses align with those in the revised version of the manuscript).

	CD47 gene deletion experiment	CD47 neutralizing experiment
Mouse	CD47^{fl/fl} , CD47^{FoxJ1} , CD47^{LysM}	C57BL/6 WT
Virus concentration	100 PFU	10 PFU
Bacteria concentration	1 x 10 ⁸ CFU	5 x 10 ⁵ CFU
Related Figures	Fig. 4, Fig. 6, and Fig. S10a-j (in the revised version of the manuscript)	Fig. 5 and Fig. S10k-m (in the revised version of the manuscript)
References	#1 , #2 , #3 , #4	#5

As indicated in the table above, we employed two distinct concentrations of virus and bacteria in the CD47 gene deletion experiment and the CD47 neutralizing experiment. We employed two distinct concentrations for two main reasons: *i*) Firstly, we utilized two different kinds of mice: Our in-house mice (*CD47^{fl/fl}*, *CD47^{FoxJ1}*, and *CD47^{LysM}*) and commercially purchased C57BL/6 WT mice, which exhibited higher sensitivity to the viral infection than the in-house mice. *ii*) Secondly, the neutralizing effect of the α -mCD47 antibody was only evident when lower concentrations of virus and bacteria were employed to C57BL/6 WT mice. This led us to hypothesize that the total quantity of administered nasal antibodies might not have been sufficient to counter the higher concentrations of virus and bacteria effectively.

As depicted in the figure below, it's important to emphasize that Fig S9 in the initial submission (*left panel*) was not a suitable control experiment. This error was a result of our utilization of higher concentrations of virus and bacteria in the experiment involving C57BL/6 WT mice. Consequently, we have replaced it with a new Supplementary Figure S10 (*right panel*), in which higher concentrations of virus and bacteria were employed, this time with *CD47^{fl/fl}* mice as the subjects. Our findings, based on the observed

body weight loss in C57BL/6 mice and $CD47^{fl/fl}$ mice following viral infections (77.5% vs. 84.8%, respectively), conclusively demonstrated that C57BL/6 WT mice exhibited a higher sensitivity to viral infection compared to the in-house mice ($CD47^{fl/fl}$). However, it's worth noting that in both scenarios, viral or bacterial single infections did not result in fatalities on their own, and significant survival differences were evident between single and double infections. Please refer to our response to Reviewer #3's comment for information on mouse control experiment [Response 3-1].

A crucial aspect that we overlooked pertains to animal ethics. We failed to adhere to the guideline stipulating that mice experiencing significant morbidity or exhibiting more than a 30% weight loss should be humanely euthanized and regarded as having succumbed to the infection (Ref #6). We have incorporated this content into both the Results section (lines 329-331) and the Methods section (lines 576-578). Accordingly, following the guidelines, we excluded mice from Figure 4 of the initial submission [Seven $CD47^{fl/fl}$ mice in Fig. 4a & 4b (*left upper panel* in the figure below), and three IgG2a-treated mice and three α -mCD47 Ab-treated mice in Fig. 4j, k (*left lower panel*)], and have subsequently conducted all mouse super-infection models following this correction. In all % body weight data in the revised version of the manuscript (Fig. 4a, Fig. 4j, Fig. 5a, Fig. 6a, Fig. S8a, Fig. S9d, Fig. S10b, and Fig. S10l), you can observe the dotted lines indicating the body weight exclusion cut-off. Reviewer #3 also raised a similar concern; please refer to our response [Response 3-1] for more information on this matter.

2. What leads from bacterial adhesion to barrier loss? The authors show that ZO-1 protein levels and distribution are perturbed by virus infection, for many researchers this alone would be sufficient (and maybe superior) to explain increased bacterial burden – as this defines proper barrier function. In contrast to this, the mechanism described here should lead to accumulation of bacteria without explaining how they actually breach the barrier more efficiently. However, in the mouse experiments, bacterial burden is unaffected upon treatment (fig. 5e, n). Even if the authors added 10 more mice per group and reached statistical significance, the less than 2 fold difference in bacterial load is much less than what described in other models of coinfection where often several logs difference are found. Therefore, the whole narrative is not quite convincing for me.

[Response 1-2] We acknowledge the reviewers' comment and appreciate their insights into our research. In response to the query regarding the mechanism that leads from bacterial adhesion to barrier loss, we propose that the disruption of epithelial barrier integrity following *S. aureus* infection necessitates not only ZO-1 disconnection but also CD47 induction and its exposure to the apical side of ciliated cells, providing an attachment site for the bacteria. This attachment may give bacteria the opportunity to proliferate and potentially breach the barrier, as illustrated in the figure below (Fig. 6j in the revised version of the manuscript). We observed that while viral infection did perturb ZO-1 protein levels and its distribution, it became evident that neither viral nor bacterial single infections facilitate the penetration of FITC-dextran nor did they affect trans epithelial electrical resistance (TEER). These observations underscore the complexity of the mechanisms underlying bacterial penetration and barrier loss.

Figure 6 in the revised version

During our revision process, we have enhanced the significance of the data related to the bacterial burden in the lung (Fig. 4e and Fig. 5e in the figure below). In response to the feedback from reviewer #3, we acknowledged the importance of assessing bacterial burden not only in the lung but also in the spleen. Remarkably, we identified a significant improvement in the bacterial burden in the spleen in $CD47^{FoxJ1}$ mice (Fig. 4f) and the α -mCD47 antibody-treated mice (Fig. 5f), in comparison to $CD47^{fl/fl}$ mice and the IgG2a-treated mice, respectively. This finding is consistent with the reviewer's concern and indicates that the bacterial burden in the lung might not entirely account for the morbidity in super-infection, as it was only marginally affected upon treatment (about 2-fold difference). As reviewer #3 pointed out, the presence of bacteremia can be considered as evidence of invasion and is a critical factor contributing to mortality (Ref #7). In light of this, we have included the following sentences in the Discussion section (lines 441-452),

highlighting the need for further investigation to elucidate how the accumulated bacteria breach the barrier. We hope these additional details address the reviewer's concerns and enhance the comprehensibility of our research: "To establish an *in vivo* correlation with the observed loss of the epithelial barrier function in our *in vitro* super-infection model (Fig. S5 and Fig. 2), we quantified bacterial CFU in the spleen as an indicator of invasion. Our findings demonstrated that the absence of CD47 in ciliated cells (Fig. 4f) or the absence of FnBP in *S. aureus* (Fig. 6f) indeed provided protection against *S. aureus* invasion. Notably, this protective effect was not observed in myeloid cell-specific knockout mice (Fig. 4o). Additionally, the use of anti-CD47 neutralizing antibody treatment proved effective in preventing bacteremia (Fig. 5f). Nonetheless, we have not yet been able to elucidate the precise mechanisms by which the accumulated bacteria breach the barrier more effectively. Further research aimed at understanding how *S. aureus* interacts with CD47 to facilitate its invasion process remains a valuable and ongoing area of investigation."

3. There is a woeful lack of quantification in this paper when histology and WB data is shown: fig.s 1a, 1b, 1e, 2b, 3c, S8, to name just the first that spring to mind. Without more serious quantification using biological replicates, the data shown here is too anecdotal.

[Response 1-3] We have taken the matter of quantification very seriously and have provided detailed quantification for all histology and Western blot data, both pre-existing and newly added (Fig. 1a, Fig. 1b, Fig. 1e, Fig. 2b, Fig. 3c, Fig. S2a, Fig. S2b, Fig. S2d, Fig. S4d, Fig. S4h, Fig. S5b, Fig. S5g, and Fig. S8e in the revised version of the manuscript). This quantification, conducted with biological replicates, enhances the robustness and reliability of the data presented in our paper. We are committed to ensuring the scientific rigor of our findings and sincerely appreciate the reviewer's diligence in this regard.

4. Fig. S1, Proteomics: ALI cultures are mixed cultures, how do the authors know that protein changes are not due to changes in cell representation?

[Response 1-4] Reviewer #1's observation is indeed valid. In our proteomics data, we cannot definitively ascertain whether the observed protein changes are solely a result of changes in cell representation. To address this concern, we conducted cytospin immunofluorescence staining following viral infection (*right panel* in the figure below). In comparison to our previously published data, where the status was non-viral infection (*left panel*, Ref #8), we now have verified that the composition of cells remained consistent following viral infection. Specifically, we observed that the proportion of ciliated cells (12.6% vs. 8.7%), secretory cells (15.6% vs. 11.6%), and basal cells (18.2% vs. 16.7%) exhibited minimal changes. However, a notable increase in CD47 expression was observed following viral infection (white arrow, *right panel* in the figure below). This finding underscores that the changes in protein expression, particularly in the case of CD47, are not solely attributed to changes in cell representation.

HNECs in our previous study (Ref#8)	Cytospin IF staining of HBECs at 1 dpi (Figure for response)
[REDACTED]	  Influenza NP   Ciliated cells 8.7%   Secretory cells 11.6%   Basal cells 16.7%    + Influenza virus   Influenza NP CD47 Influenza NP CD47 Ac-α-1-tubulin MUC5AC p63 

5. Fig. S2: How can the FC of CD47 be 1.15 (as stated in the text) and yet the WB and microscopy shows huge differences, certainly more than a 1.15 increase?

[Response 1-5] In the initial stages of this study, we adopted a proteomics approach to identify membrane proteins whose expression is triggered by viral infection. Our hypothesis was based on the idea that substantial alterations in protein expression might become evident around 5 days post-infection (dpi). Given our use of human cells, our expectation was that the cellular response would closely parallel the clinical aspects (Ref #9). As we validated our proteomics data through Western blotting, we observed intriguing dynamics in CD47 expression. In HNECs, CD47 expression peaked at 1 dpi and gradually decreased, with a marginal increase at 5 dpi (*left panel* in the figure below). This explains the marginal nature of our proteomics result (fold change 1.15 at 5 dpi) and the substantial difference observed in the Western blot data (at 1 dpi). Following this discovery, we shifted our focus to the examination of CD47 expression in human airway epithelial cells at 1 dpi. Conversely, in mice, CD47 expression displayed a gradual increase and reached its peak at 5 dpi (Fig. S8d, *right panel*). These findings emphasize the importance of considering the specific context and timeframe of protein expression changes in response to viral infection. This revised response provides a more comprehensive explanation of the observed differences in CD47 expression dynamics between primary airway epithelial cells and mice.

6. The NFκB blockade could block CD47 indirectly: IFN induction is NFκB-dependent, is CD47 induction IFN dependent? Use IFNAR/IFNLR blocking Abs.

[Response 1-6] We have conducted an extensive investigation to determine whether CD47 induction is dependent on interferon (IFN). We observed an increase in CD47 expression following viral infection (Fig. S3c, d), and this effect was replicated in HBECs treated with recombinant human IFN-β (Fig. S3e) and IFN-λ1 (Fig. S3f). Notably, the induction of CD47 by the virus was entirely suppressed when anti-IFNAR (Fig.

S3c) or anti-IFNLR (Fig. S3d) neutralizing antibodies were applied. Additionally, the induction of CD47 by recombinant human IFN- β was effectively inhibited by anti-IFNAR neutralizing antibodies (Fig. S3e), while CD47 induction by recombinant human IFN- λ 1 was specifically blocked by anti-IFNLR neutralizing antibodies (Fig. S3f). These results collectively demonstrate that CD47 induction occurs via the NF- κ B/IFN pathway. We have included this content within the Results section of our manuscript (lines 158-168).

7. Not all cells within the epithelial cultures express CD47, are those who do virus-infected? Double staining for viral protein and CD47.

[Response 1-7] To address the reviewer's concern, we performed double staining on virus-infected HBECs to assess the co-localization of CD47 and the viral protein NP, as depicted in the figure below. Interestingly, our observations revealed that some CD47-positive cells (which are primarily expected to be ciliated cells based on Fig. 1e and Fig. S2d) indeed co-localized with viral protein expression. This indicates the presence of virus-infected ciliated cells. However, it's noteworthy that subsequent observations showed cells that were CD47-positive but not IAV-positive, suggesting the existence of non-infected ciliated cells, and infected cells that were IAV-positive but not CD47-positive, indicating infected non-ciliated cells. In summary, our findings suggest that virus-infected ciliated cells can induce CD47, while neighboring infected cells induce CD47 in ciliated cells. The precise mechanism by which neighboring infected cells induce CD47 in ciliated cells remains unclear at this stage. In acknowledgment of this issue, we have incorporated sentences in the Discussion section to further highlight this observation (lines 396-405): "In human tissues, influenza viruses primarily bind to ciliated cells, whereas in differentiated airway epithelial cell cultures, they infect non-ciliated cells and, to a lesser extent, ciliated cells. Recent studies have indicated that SARS-CoV-2 also exhibit a preference for targeting ciliated cells and interferes with mucociliary clearance. Consistently, our observations indicated that not all CD47-positive cells completely overlapped with viral protein expression. It's important to note that while CD47 induction by viral infection is specific to ciliated cells (Fig. 1e and Fig. S2d), not all ciliated cells were infected (data not shown). The precise mechanism by which neighboring infected cells induce CD47 in ciliated cells remains unclear at this stage."

8. Text reference to fig. 2e, accepted m/s: does this mean that the data is not yet publicly available?

[Response 1-8] The manuscript has indeed been accepted, and it is cited as reference #43 (Hyung-Ju Cho, *et al.* IL-4 drastically decreases deuterosomal and multiciliated cells via alteration in progenitor cell differentiation. *Allergy*. 2023 Jul;78(7):1866-1877. doi: 10.1111/all.15705. Epub 2023 Mar 18.).

9. Fig. S4: In our own experience, when cultures really are as they look in S4d and h, then the TEER is zero or near zero. Here, huge holes appear to be visible in the culture which would lead to a complete collapse of the resistance. Also, are these cultures done without antibiotics? The bacteria must take over completely by d5 and even more so by d7 which would perturb the tissue culture system to a point where cell death will occur that is not representative of physiological processes. In this figure, the *p* values of double infected vs single infected are important as the argument is that there is a synergy, so comparison to mock alone is insufficient.

[Response 1-9] We appreciate the reviewer's valuable input and would like to provide clarification on the TEER measurements and the bacterial culture conditions used in our experiments.

As stated in the Results section (lines 207-212) and indicated in Fig. S4, we indeed measured TEER at 3 dpi and showed the terminal effects of super-infection at 7 dpi for HNECs and 5 dpi for HBECs. It is important to note that we did not measure TEER at 7 dpi or 5 dpi, and the results reflect the effects of earlier time points on TEER.

In our experimental setting, following bacterial infection, we washed the bacteria right after the infection and cultured the cells without antibiotics, as described in the Method section (lines 562-564).

Regarding the statistical analysis, we agree that the *p*-values are important for demonstrating the synergy between double infected and single infected conditions. We have performed this comparison and presented it in Fig. S4a, Fig. S4b, Fig. S4e, and Fig. S4f in the revised version of the manuscript. These results directly address the question of whether the combined effect of double infection significantly differs from the individual single infections, providing further support for our argument.

10. How do you explain the discrepancy between 2a, b and c? There is an only twofold reduction in mRNA which however leads to disappearance of protein in the WB (but not quantification, *n*=1?), and a massive increase in permeability. Similar, S5A, C?

[Response 1-10] We would like to clarify the observed discrepancy between the data presented in Fig. 2a, b, and c. It's important to note that we conducted quantitative analysis of all Western blot bands for accurate quantification, as illustrated in Fig. 2b (*n* = 5) and Fig. S5b (*n* = 5). This analysis revealed that, in line with the mRNA data, there is an approximately two-fold reduction in CD47 protein expression. As a result, the massive increase in permeability observed following super-infection was also reduced by approximately two-fold.

11. Throughout the study, both binding via CD47 and the involvement of fibronectin binding protein is shown only indirectly. At no point is direct interaction between bacteria and CD47 or pull downs of CD47 by the bacteria shown. Similarly, using FnBP mutants to show the dependency on *fn* is an elegant but inconclusive way to show involvement of *fn* or indeed the binding protein – these mutants may have additional defects, leading indirectly to the phenotypes shown here. Along these lines, fig. 3h shows a laudable effect that however needs another orthogonal control, for instance another, unrelated protein.

[Response 1-11] We have taken note of the reviewer's concern regarding the direct demonstration of the interaction between bacteria and CD47, as well as the potential indirect effects of fibronectin binding protein (FnBP) mutants. In response, we conducted *in vitro* pull-down assays using *S. aureus* mutant strains in which other unrelated proteins were deleted. These proteins include clumping factor A (*clfA*), serine-aspartate repeat protein E (*sdrE*), iron-regulated surface protein B (*isdB*), *S. aureus* surface protein G (*sasG*), as well as FnBP B (*fnbB*). This approach provides an orthogonal control by demonstrating that the observed effects are specific to the FnBP mutants and not influenced by other unrelated protein deletions.

We have included these results as Supplementary Figure 7d and e.

Minor concerns:

1. *Mx1 and Mx2 are not supposed to be membrane-bound... nuclear or cytosolic*

[Response 1-12] We agree with the reviewer's comment, and we have removed them from the sentence.

2. *line 105 CEACAM typo*

[Response 1-13] We have corrected it to CEACAM5.

3. *line 135 where do you show us that this is on the apical side?*

[Response 1-14] Reviewer #1 is indeed correct, and we appreciate this valuable observation. To address this point, we have included the following sentence in the Results section (lines 174-177): "It's worth noting that ciliated cells are a part of the epithelium, lining the apical surface of the respiratory tract and exposing their cilia to the lumen. Therefore, the detection of CD47 co-expressed with Ac- α -tubulin indicates that it is present on the apical side of the ciliated cells."

4. *Fig. S4G, so most cells are dead? How do you translate LDH levels into % viability?*

[Response 1-15] No, most cells are alive. We have corrected the label to "% cell death" to accurately reflect the data (Fig. S4c, g). We apologize for the initial incorrect expression.

5. *Figure S6, the authors need to show single infection with control shRNA.*

[Response 1-16] We have performed additional experiments of single infection with control shRNA and added new Supplementary Figure 6a and Supplementary Figure 6b to address this concern.

6. *line 189 and later, e.g. 211: The authors need to explain in the main text how bacterial adhesion is measured.*

[Response 1-17] We have described how bacterial adhesion was measured in the Results section (lines 246-247): "To validate this hypothesis, we initially performed a bacterial adhesion assay to quantify the number of bacteria adhering to HBECs after a 3-hour co-incubation".

7. *In line 189 and 198 the authors cannot say here that this is adherence to CD47 as they did not test this here. It is adherence to the cells, and it is CD47 dependent which may be direct or indirect at this point.*

[Response 1-18] We appreciate the reviewer's insight, and we have revised the sentences in question accordingly (lines 247-251): "This analysis revealed a significant increase in the adherence of *S. aureus* to the cells following viral infection in HBECs (Fig. 3d). Additionally, the virus-induced bacterial adherence was

inhibited when the cells were pre-incubated with α -hCD47 antibodies (Fig. 3d), suggesting that this cell-to-bacteria interaction may depend on CD47."

8. Fig. S10a,b, what is shown here? The M and f nomenclature is not in text or legend. Why is the CD47 band identical in b in the presence or absence of Cre?

[Response 1-19] We apologize for the oversight in our presentation. In Supplementary Figure 10a and Supplementary Figure 10b from the initial submission, we intended to illustrate the results of our experiments using male (M) and female (F) mice. We have updated the legend for Supplementary Figure 9 in the revised version of the manuscript to clarify this nomenclature.

Regarding the identical band in Supplementary Figure 10b (Fig. S9b in the revised version of the manuscript), it is not the CD47 band; it represents the presence of a floxed cassette in the mouse genome. To avoid any further ambiguity, we have revised the figure and accompanying text to provide a more comprehensive explanation.

9. Fig. S10d these virus measurements are a bit late, what about d2,3,4 p.i.?

[Response 1-20] We have measured virus titer at 2, 3, and 4 dpi in addition to 7 dpi. However, no significant difference was observed in viral titer between $CD47^{ff}$, $CD47^{FoxJ1}$ and $CD47^{LysM}$ mice at these earlier time points (Fig. S9j and Fig. S9k).

Reviewer #2 (Remarks to the Author):

Summary – The manuscript by Moon and colleagues identifies a novel role for influenza-induced epithelial CD47 in promoting attachment of S. aureus in vitro and in mediating severe super-infection disease in vivo. The authors show that influenza induces CD47 in human nasal and bronchial ciliated epithelial cells via NF- κ B induction. CD47 is then shown to be required for in vitro super-infection induced epithelial barrier leak and loss of TEER. The authors then demonstrate that S. aureus FnBP plays a contributing role to S. aureus adherence via interaction with CD47. Finally, in the mouse model, epithelial specific deletion of CD47 results in decreased super-infection disease phenotype and a modest reduction in mortality. CD47 monoclonal antibody treatment mimicked these effects. Overall, this is a well designed and clearly presented study with compelling results. The manuscript could be improved as detailed below.

Major Comments –

1) Influenza-induced CD47 time course – It would be helpful if the authors could show a time course for CD47 induction during influenza in vivo. Does CD47 expression track with the window of enhanced susceptibility to secondary infection (day 5-10) post infection? This would help delineate the specific contribution of CD47 to susceptibility.

[Response 2-1] We greatly appreciate the valuable suggestion from the reviewer to conduct this insightful experiment. We have thoroughly examined CD47 mRNA levels (Fig. S8b in the revised version of the manuscript) and protein levels (Fig. S8c and Fig. S8d) in mouse lung at 2, 3, 5, 7, and 10 days post-infection (dpi). To our surprise, we observed a gradual increase in both mRNA and protein levels, with a peak at 5 dpi, followed by a subsequent decrease. This time course data provides valuable insights into how CD47 expression aligns with the window of enhanced susceptibility to secondary infection, which is typically observed between days 5-10 post-infection.

Moreover, this experiment not only helps delineate the specific contribution of CD47 to susceptibility but also provides a solid rationale for the timing of antibody administration. We administered antibodies twice, at 5 dpi and 7 dpi, precisely when CD47 induction by viral infection is at its peak, thus offering an opportune moment for CD47 inhibition.

2) *S. aureus* mutants – It would be great to close the pathway loop by infecting WT and Foxj1-Cre CD47^{fl/fl} mice with the *S. aureus* lacking FnbA/B to determine if the phenotype matches to the expected model. For example in WT the mutant *S. aureus* should have decreased pathogenicity, while in the epithelial CD47 KO there should be no effect of deletion of FnbA/B. It would be helpful to also cite Groud et al. *Microbiol Spectrum* 2022, which tested FnbB deletion in a similar super-infection model.

[Response 2-2] We would like to express our gratitude to the reviewer for this invaluable suggestion. In order to determine if the phenotype matches with the expected model, we conducted experiments involving CD47^{fl/fl} (WT) and CD47^{FoxJ1} (cKO) mice infected with *S. aureus* strains with and without FnBP A/B (FnBP A+/B+ and FnBP A-/B-, respectively). Remarkably, the mutant *S. aureus* (FnBP A-/B-) exhibited decreased pathogenicity in WT mice, while there was no discernible effect of FnBP A/B deletion in the epithelial CD47 cKO mice (Fig. 6b-i), suggesting that the interaction between airway epithelial CD47 and *S. aureus* FnBP is critical for the development of viral–bacterial super-infection. We have included the following sentence in the Results section (lines 367-374): To fully confirm the interaction between FnBP and CD47 *in vivo*, we infected both CD47^{fl/fl} and CD47^{FoxJ1} mice with two *S. aureus* strains, FnBP A+/B+ and FnBP A-/B-, to verify whether the observed phenotype aligns with our expected model. As expected, FnBP A-/B- exhibited reduced pathogenicity when infecting CD47^{fl/fl} mice, similar to the effect of FnBP A+/B+ in CD47^{FoxJ1} mice, while there was no observable effect when either FnBP A+/B+ or FnBP A-/B- infected CD47^{FoxJ1} mice (Fig. 6a-i). Taken together, our results demonstrate that both airway epithelial CD47 and *S. aureus* FnBP are essential for the development of viral–bacterial super-infection (Fig. 6j).

The paper by Groud et al. has been cited as reference #49 to provide relevant context for this experiment (Jennifer A Groud, et al. Novel Requirement for Staphylococcal Cell Wall-Anchored Protein SasD in Pulmonary Infection. *Microbiol Spectr.* 2022 Oct 26;10(5):e0164522. doi: 10.1128/spectrum.01645-22. Epub 2022 Aug 30.).

Minor Comments –

1) Line 308 – Clara Cells have been renamed to Club Cells.

[Response 2-3] As per the request from reviewer #4, we have decided to remove the experimental results of SARS-CoV-2 and accompanying text, including any reference to Clara cells.

Reviewer #3 (Remarks to the Author):

To the authors

The authors have performed an extensive analysis of a potential mechanism involved in *S. aureus* superinfection following flu, highlighted with the use of both FoxJ1 and CD47 LysM Cre floxed mice. While the *in vitro* data appear straightforward, suggesting the upregulation of CD47 in epithelial cells following influenza infection and recognition of this by *S. aureus* FNBP, the mouse models do not clearly support the underlying hypothesis. The expected WT mouse control is only provided in a supplemental figure - but not with the mouse models using Ab or floxed mice. Additional controls are necessary to adequately

interpret the data shown.

1- WT controls are needed in the mouse studies. In Fig 4 - There appears to be a substantial survival effect (most mice are alive 4-5 days post *S. aureus* infection) with the administration of either the control Ab or that targeting CD47 - if one compares survival at 24 hours post Staph infection- which is a typical outcome point. This is very different than the control data showing that virtually all of the mice are dead within 24 hours following the administration of *S. aureus*. As survival is the outcome in the Supp. Fig 9.

WT mouse infection experiments using PBS as well as antibody (which non-specifically binds *S. aureus* protein A) are needed in parallel.

In Fig 4B - how many mice survived beyond day 10? It appears that the conclusions that the *FoxJ1* floxed mice have increased survival may be based on a very small number of survivors, but this is impossible to tell from the data presented. Of note, the Suppl fig11 data indicates that CD47 is critical in myeloid cells - as these mice all died apparently within 24 hours post *S. aureus* infection - but this is impossible to judge without a WT mouse control is necessary. Again, mouse numbers are not provided for the time points post infection. While changes in mouse weight are outcomes for viral infection, cfus in addition to death, are typically included in bacterial models.

Previous studies (Nauseef 2014 JI) demonstrated that CD47 functions as an "eat me" signal that is critical for macrophage clearance of neutrophils that have ingested *S. aureus*. In the absence of CD47- there is increased infection. Thus, the relative roles of CD47 on epithelial versus myeloid cells may be very distinct and needs to be better addressed.

[Response 3-1] We appreciate the reviewer's valuable feedback and have made the following modifications and additions to address the concerns raised:

Control Experiments: To ensure robust comparisons, we have conducted control experiments using two distinct sets of mice, as illustrated in the figure below. We used *CD47^{fl/fl}* mice for the CD47 gene deletion experiment (*top panel* in the figure below), and C57BL/6 WT mice for the CD47 neutralizing experiment (*bottom panel*). This approach ensures that we have appropriate control groups to evaluate the impact of our experimental interventions.

Figure S10 in the revised version

As previously addressed in [Response 1-1], we acknowledge discrepancies in Supplementary Figure 9 from the initial submission, primarily stemming from fluctuations in virus and bacteria concentrations during experiments involving C57BL/6 WT mice. To rectify this, we have introduced a revised Supplementary Figure 10b (*left panel* in the figure below), demonstrating experiments using higher concentrations of virus and bacteria with *CD47^{ff}* control mice. Additionally, we have included another control experiment in Supplementary Figure 10l (*right panel*), in which C57BL/6 WT mice were used. In alignment with our explanation in [Response 1-1], it is worth noting that C57BL/6 WT mice are generally more susceptible to viral infections compared to our in-house *CD47^{ff}* mice. However, due to our deliberate use of reduced virus and bacteria concentrations in the C57BL/6 WT mouse experiments, both groups of mice exhibited similar changes in % body weight and % survival rates (Fig. S10b and Fig. S10l). Please refer to our response to Reviewer #1's initial comment for information on virus or bacteria concentrations and exclusion criteria [Response 1-1].

In reference to Figure 4k from the initial submission, we sincerely apologize for the significant oversight in indicating the incorrect time point (5 dpi) for *S. aureus* infection (*left panel* in the figure below). In the revised manuscript (*right panel*), we have rectified this error by adjusting the arrow to accurately denote the timing of *S. aureus* infection (7 dpi). This correction reflects that approximately half of the mice died within the 5-day period, spanning from day 7 to day 12 following *S. aureus* infection, under the administration of either the control antibody or the CD47-targeting antibody.

Number of survived mice: In response to the reviewer's comment, we have included the number of mice in all % survival data (Fig. 4b, Fig. 4k, Fig. 5b, Fig. 6b, Fig. S10b, and Fig. S10l), addressing the concern about the number of survivors beyond day 10. Specifically, ten out of 22 $CD47^{FoxJ1}$ mice survived following super-infection (45.5%), while none of 35 $CD47^{fl/fl}$ mice survived (0%) (*left panel* in the figure below). Based on this data, we have concluded that $CD47^{FoxJ1}$ mice have increased survival compared to $CD47^{fl/fl}$ mice.

Bacterial CFU: In response to the reviewer's feedback, we recognize the importance of assessing bacterial burden not only in the lung but also in the spleen. To address this, we have included the assessment of bacterial burden in the spleen (Fig. 4f and Fig. 5f) in addition to the data presented for the lung (Fig. 4e and Fig. 5e) in the revised version of the manuscript.

CD47 in myeloid cells: Regarding the Supplementary Figure 11 from the initial submission data, we have shown that $CD47^{LysM}$ mice did not exhibit a protective effect against super-infection (*right panel* in the above figure). However, based on the literature the reviewer referred to, we have thoughtfully included the following sentence in the discussion on page 18 (lines 460-465): “Previous studies have demonstrated that CD47 functions as an “eat me” signal critical for macrophage clearance of neutrophils that have ingested *S. aureus*. In the absence of CD47, there is an increased risk of infection. Therefore, it is possible that the roles of CD47 on epithelial cells versus myeloid cells are distinct, and further investigation is required to fully understand these differences.” (Ref #10).

These adjustments and additions improve the quality and reliability of our data, address the need for appropriate control experiments, and enhance the interpretability of the results.

2- Characterization of the cell lines - A major point of the manuscript is the expression of CD47 by ciliated epithelial cells - thus the use of polarized, presumably ciliated cell lines is expected. The authors provide an extensive analysis of nasal epithelial cells, a site of colonization (.i.e.no immune response?) but not infection. In the suppl fig 7 CD47 appears to be predominantly expressed in "deuterosomal" cells -but is co-expressed with FoxJ1 to a lesser extent in multi-ciliated cells. As the nasal epithelium is not a site of *S. aureus* infection - is there a similar analysis of the human bronchial epithelial cells? What was their expression of cilia? Of CD47 and FoxJ1?

[Response 3-2-1] In response to the reviewer's query regarding the analysis of human bronchial epithelial cells, including their cilia expression, CD47, and FoxJ1 levels, we have conducted a comprehensive examination. We have analyzed three publicly available scRNA-seq datasets performed using HBECS (Ref #11, #12, and #13) and have presented a summary of our findings in the accompanying figure.

Except for the first dataset, where the total analyzed cell number is relatively small (about one-tenth of the other datasets), our analysis of two scRNA-seq datasets reveals that CD47 appears to be predominantly expressed in "deuterosomal" cells, but it is also co-expressed with FoxJ1 to a lesser extent in multi-ciliated cells at a site of infection. This result supports a major point of the manuscript that the

expression of CD47 is specifically present in ciliated epithelial cells. We have included the following sentence in the Result section (lines 180-184): "To validate these findings in HBECs, we analyzed two publicly available scRNA-seq datasets performed with HBECs. Our analysis revealed that CD47 expression is predominantly expressed in FoxJ1⁺ cells, encompassing deuterosomal cells, and to a lesser extent, multi-ciliated cells within HBECs (data not shown).

These are important - since A549 cells, as a non-polarized carcinoma cell line are not especially relevant to the airway.

[Response 3-2-2] We acknowledge the reviewer's insightful comment regarding the relevance of the A549 cell line, which indeed lacks polarization and does not adequately represent the complexities of the airway. In response to this valuable comment, we have carefully reviewed our study and made substantial revisions. Notably, we have replaced all data in Figure 3, previously derived from A549 cells, with data acquired from human bronchial epithelial cells (HBECs). Importantly, our findings from both A549 and HBECs exhibited compatibility, reinforcing the robustness of our results.

Why are viable S. aureus necessary to induce CD47? Do they require uptake and internalization? This seems distinct from a simple ligand-receptor interaction.

[Response 3-2-3] We appreciate the reviewer's valuable comment. Our hypothesis is that viable *S. aureus* may rely on the apically exposed CD47, induced by viral infection, as an attachment site for their proliferation and subsequent barrier breach. However, in this study, distinguishing between uptake and internalization in primary epithelial cells presented technical challenges. For a more in-depth exploration of these challenges, we refer the reviewer to our response to Reviewer #4's comment **[Response 4-5]**.

We acknowledge that the interaction between *S. aureus* and CD47 goes beyond a simple ligand-receptor interaction. During the revision process, we have bolstered our findings with additional evidence supporting the direct binding of *S. aureus* to HBECs through a specific interaction between FnBP and CD47 (Fig. S7d and Fig. 6 in the revised version of the manuscript). Nonetheless, we concur that further studies will be necessary to fully elucidate the precise mechanisms of the interaction between *S. aureus* and CD47.

3- Several experiments demonstrate the loss of the epithelial barrier function in vitro. As an in vivo correlate - what were the rates of S. aureus bacteremia as evidence of invasion? Did the lack of CD47 protect from in vivo invasion? Was this the case with the myeloid specific KOS as well as the FoxJ1 floxed mice??

[Response 3-3] We appreciate the insightful comment from the reviewer. To provide an *in vivo* correlate to our *in vitro* experiments assessing the loss of epithelial barrier function, we conducted an *in vivo* bacteremia assay by quantifying bacterial colony-forming units (CFU) in the spleen, which serves as an indicator of invasion. Our findings consistently showed that the absence of CD47 in ciliated cells (Fig. 4f) indeed conferred protection against *S. aureus* invasion. However, this protective effect was not observed in myeloid cells (Fig. 4o). Additionally, the use of anti-CD47 neutralizing antibody treatment also proved effective in protecting against bacteremia (Fig. 5f). As a result of these observations, we have concluded that the loss of epithelial barrier function plays a pivotal role in driving *S. aureus* bacteremia, a significant factor contributing to super-infection *in vivo*.

The study is somewhat innovative in clarifying the crucial role that CD47 plays in viral-bacterial super-infection. And it has certain clinical application value, which provides new ideas for the treatment of super-infection. This manuscript has novelty and innovation. However, there were several main concerns that should be addressed as follows:

Major issues:

1) Introduction part of the paper is poorly cited and does not clarify why CD47 was chosen for the study. The summary of this study at the end of the introduction is not sufficiently clear.

[Response 4-1] We have rewritten the introduction to address the reviewer's concerns and provide better clarity on the rationale for selecting CD47 as the subject of our study.

1. *Why CD47 was chosen for the study* (lines 93-103): Notably, the specific involvement of CD47 in airway epithelium during super-infection remains unexplored. In a proteomics analysis of nasal epithelial cells infected with influenza virus, we observed the presence of CD47. Through the validation process, we uncovered that CD47 induced by viral infection was not only detected in the colonization site (nasal epithelium) but also at the site of infection (bronchial epithelium). Given that CD47 seems to be predominantly expressed in FoxJ1⁺ cells (deuterosomal cells and, to a lesser extent, multi-ciliated cells) based on single-cell RNA sequencing and immunostaining analysis of upper and lower airway primary epithelial cells, we initiated an investigation into the involvement of CD47 in secondary bacterial infections using a FoxJ1⁺ cell-specific CD47 gene-deletion mouse model and CD47 neutralizing antibodies.
2. *The summary of this study* (lines 103-113): In our pursuit to identify a bacterial component that interact with CD47, we conducted experiments involving five *S. aureus* mutant strains with deleted cell wall-anchored proteins. Remarkably, our investigations revealed that only fibronectin-binding proteins (FnBP) exhibited a strong affinity for CD47. To further solidify our findings, we employed a combination of FoxJ1⁺ cell-specific CD47 gene-deletion mice and a FnBP mutant strain of *S. aureus* in the context of super-infection. This approach allowed us to establish that the specific interaction between airway epithelial CD47 and bacterial FnBP plays a pivotal role in causing super-infection. By comprehensively exploring the role of CD47 in facilitating secondary bacterial infections, this study may pave the way for the development of innovative therapeutic approaches, ultimately leading to improved clinical outcomes for patients.

2) *Why is iTRAQ done 5 days after IAV infection. Please clarify the basis of timepoint selection.*

[Response 4-2] We appreciate the reviewer's insightful comment. In initiating this study, we conducted iTRAQ proteomics 5 days after IAV infection based on a specific rationale. Our selection of this time point was driven by the hypothesis that significant changes in protein expression, particularly those relevant to viral infection, might become more pronounced around 5 days post-infection (dpi). This timeframe aligns with clinical observations (Ref #9) and is well-suited to capturing relevant protein dynamics during infection.

As we validated our proteomics data through Western blotting, we observed intriguing dynamics in CD47 expression. In human nasal epithelial cells (HNECs), CD47 expression peaked at 1 dpi, gradually decreasing with a slight increase at 5 dpi (*left panel* of the figure below). Conversely, in mouse models, CD47 expression exhibited a gradual increase, reaching its peak at 5 dpi (*right panel*). These observations emphasize the significance of considering the specific context and timeframe of protein expression changes in response to viral infection. This revised response provides a more detailed explanation of the

dynamics in CD47 expression that guided our experimental timeline (Reviewer #1 also raised this concern: please refer to our response [Response 1-5] for further details.).

3) Can *S. aureus* infection alone or super-infection induce CD47 expression? This is of major importance.

[Response 4-3] We are immensely grateful to the reviewer for this invaluable suggestion. In response to the reviewer's inquiry, we conducted thorough investigations to determine whether *S. aureus* infection alone or super-infection had the capacity to induce CD47 expression. Our findings provide conclusive evidence that *S. aureus* single infection did not induce CD47 expression in both *in vitro* and *in vivo* settings. Furthermore, no synergistic effect was observed with viral infection, as evidenced in Supplementary Figure 8f and 8g.

4) Compared with wild mice, there was no difference in the viral load in the lungs of IAV infected with ciliated cell-targeted CD47-deficient mice for 7 days, but after 7 days of infection, did CD47 affect the innate immunity of mice, leading to changes in viral load, and affect the severity of lung lesions?

[Response 4-4] We have thoroughly examined the innate immunity of the virus-infected mice and have continued to monitor the viral load in the lungs of IAV-infected mice until 10 days post-infection (dpi). Our observations have revealed that there is no significant difference between *CD47^{fl/fl}* mice and *CD47^{Foxj1}* mice in terms of innate immunity (IFN- λ 2/3, Fig. S9g) and viral load (Fig. S9h and Fig. S9i). These findings suggest that, at the time points assessed, CD47 deficiency in ciliated cells does not appear to have a substantial impact on viral load or the innate immunity.

5) Fig. 3: cells were washed twice with PBS to detect bacteria. The treatment would result that the detected bacteria which included not only bacteria adhered to the cells but also internalized bacteria into cells.

[Response 4-5] We greatly appreciate the reviewer's valuable input and would like to provide an explanation regarding the technical challenges in distinguishing between uptake (bacterial adhesion) and internalization (bacterial invasion) in our experimental setup. We acknowledge the reviewer's comment that our bacterial adhesion assay cannot distinguish between bacteria adhered to the cells and those internalized within the cells. However, the detection of internally-located bacteria presents challenges within our experimental conditions. In the bacterial invasion assay, following the initial 3-hour co-incubation of cells with bacteria, any extracellular and surface-adherent bacteria are eliminated by a subsequent 2-hour incubation with media containing an antibiotics mixture (Ref #14). It is important to note that exposing the cells to bacteria for longer than 3 hours resulted in the destruction of the cell layer. As a result, primary

airway epithelial cells that were already infected by virus and further infected by bacteria beyond this 3-hour timeframe would be damaged, leading to inconclusive results.

We have incorporated the following sentence into the Discussion section of our manuscript (lines 438-452): “Due to the methodological challenges in our experimental condition, we were unable to distinctly differentiate between adhesion and invasion in this study. Instead, we focused on assessing *S. aureus* bacteremia in the context of invasion. To establish an *in vivo* correlation with the observed loss of the epithelial barrier function in our *in vitro* super-infection model (Fig. S5 and Fig. 2), we quantified bacterial CFU in the spleen as an indicator of invasion. Our findings demonstrated that the absence of CD47 in ciliated cells (Fig. 4f) or the absence of FnBP in *S. aureus* (Fig. 6f) indeed provided protection against *S. aureus* invasion. Notably, this protective effect was not observed in myeloid cell-specific knockout mice (Fig. 4o). Additionally, the use of anti-CD47 neutralizing antibody treatment proved effective in preventing bacteremia (Fig. 5f). Nonetheless, we have not yet been able to elucidate the precise mechanisms by which the accumulated bacteria breach the barrier more effectively. Further research aimed at understanding how *S. aureus* interacts with CD47 to facilitate its invasion process remains a valuable and ongoing area of investigation.”

6) Line 137-138: It would be better to modify this small title to “Epithelial CD47 in HBECs and HNECs facilitates to the super-infection”, etc., This would be in line with the Line 172-174. Knock-out and neutralization tests are only further validation.

[Response 4-6] We appreciate your suggestion, and we have modified the subheading as you recommended to read “Epithelial CD47 in HBECs and HNECs facilitates super-infection.” (line 193).

7) Line 186-187 and Fig.3: Why was A549 cells used to detect bacterial adhesion? All the previous studies used HNEC and HBEC, and did not study the expression of CD47 in A549. Different cell lines may have an effect on the results, the author should verify the universality of the results.

[Response 4-7] We concur with your observation and have taken the necessary steps to address this concern. We have replaced all data in Figure 3, obtained using A549 cells with data from HBECs to maintain consistency with the previous studies conducted in HNECs and HBECs. Thank you for highlighting this issue.

8) Supplementary Fig. 9: Did the authors measure the viral titers except for the bacteria burden?

[Response 4-8] In response to the reviewer's comment, “we have conducted a comprehensive evaluation of innate immunity and continuously monitored viral loads in the lungs of IAV-infected mice up to 10 days post-infection (dpi). Our observations revealed that there were no significant differences in viral titers between CD47^{ff}, CD47^{Foxj1}, and CD47^{LysM} mice at 2, 3, 4, and 7 dpi (Fig. S9j, k), as well as in IFN-λ2/3 levels (Fig. S9g) and viral loads (Fig. S9h, i) between CD47^{ff} mice and CD47^{Foxj1} mice at 10 dpi. These findings suggest that, at the time points examined, CD47 deficiency in ciliated cells or myeloid cells does not seem to have a substantial impact on innate immunity.” We have incorporated this valuable information into the Discussion section of our manuscript (lines 453-460).

9) Line 304-315: This paper mainly focused on influenza virus, so it is not necessary to include the experimental results of SARS-CoV-2. The description of SARS-CoV-2 has been added in the first part of

discussion, so there is no need to use another paragraph to elaborate.

[Response 4-9] We appreciate your suggestion and have accordingly removed the experimental results of SARS-CoV-2 as well as the additional paragraph, as it is not the primary focus of our paper. This adjustment streamlines the content to maintain a more focused discussion.

Minor issues:

1) Supplementary Fig. 9a: Significance analysis is recommended.

[Response 4-10] We have performed a significance analysis using a two-way ANOVA for the data in Supplementary Fig. 8a, Supplementary Fig. 9d, Supplementary Fig. 10b, and Supplementary Fig. 10l in the revised manuscript, as recommended. This additional analysis provides a more comprehensive evaluation of the presented results.

2) Supplementary Fig. 9c, Supplementary Fig. 11c, Fig. 4c: The lesion location was not shown in figures.

[Response 4-11] We have addressed this concern by adding dotted lines and arrows to indicate lymphocytic infiltration and alveolar hemorrhage, respectively, in Figure 4c, l, Figure 5c, Figure 6c, and Supplementary Fig. 10d in the revised manuscript. Thank you for your observation, and we hope this enhancement improves the clarity of our figures.

3) Line 260: "Figure. S10A" should be modified to "Supplementary Fig. 10a".

[Response 4-12] We have corrected this reference as suggested. Thank you for bringing it to our attention.

References

1. Vincent C Tam, *et al.* PPAR α exacerbates necroptosis, leading to increased mortality in postinfluenza bacterial superinfection. *Proc Natl Acad Sci U S A.* 2020 Jul 7;117(27):15789-15798. doi: 10.1073/pnas.2006343117. Epub 2020 Jun 24.B
2. Keven M Robinson, *et al.* The inflammasome potentiates influenza/Staphylococcus aureus superinfection in mice. *JCI Insight.* 2018 Apr 5;3(7):e97470. doi: 10.1172/jci.insight.97470.
3. Keer Sun, *et al.* Nox2-derived oxidative stress results in inefficacy of antibiotics against post-influenza S. aureus pneumonia. *J Exp Med.* 2016 Aug 22;213(9):1851-64. doi: 10.1084/jem.20150514. Epub 2016 Aug 15.
4. Cherrie-Lee Small, *et al.* Influenza infection leads to increased susceptibility to subsequent bacterial superinfection by impairing NK cell responses in the lung. *J Immunol.* 2010 Feb 15;184(4):2048-56. doi: 10.4049/jimmunol.0902772. Epub 2010 Jan 18.
5. Iris K Pang, *et al.* Efficient influenza A virus replication in the respiratory tract requires signals from TLR7 and RIG-I. *Proc Natl Acad Sci U S A.* 2013 Aug 20;110(34):13910-5. doi: 10.1073/pnas.1303275110. Epub 2013 Aug 5.
6. Jun Zhi Er, *et al.* Loss of T-bet confers survival advantage to influenza-bacterial superinfection. *EMBO J.* 2019 Jan 3;38(1):e99176. doi: 10.15252/embj.201899176. Epub 2018 Oct 15.
7. Christophe Langouët-Astrié, *et al.* The influenza-injured lung microenvironment promotes MRSA virulence, contributing to severe secondary bacterial pneumonia. *Cell Rep.* 2022 Nov 29;41(9):111721. doi: 10.1016/j.celrep.2022.111721.
8. Youn Wook Chung, *et al.* Apolipoprotein E and Periostin Are Potential Biomarkers of Nasal Mucosal Inflammation. A Parallel Approach of In Vitro and In Vivo Secretomes. *Am J Respir Cell Mol Biol.* 2020 Jan;62(1):23-34. doi: 10.1165/rcmb.2018-0248OC.
9. Christin Peteranderl, *et al.* Human Influenza Virus Infections. *Semin Respir Crit Care Med.* 2016 Aug;37(4):487-500. doi: 10.1055/s-0036-1584801. Epub 2016 Aug 3.
10. Mallary C Greenlee-Wacker, *et al.* Phagocytosis of Staphylococcus aureus by human neutrophils prevents macrophage efferocytosis and induces programmed necrosis. *J Immunol.* 2014 May 15;192(10):4709-17. doi: 10.4049/jimmunol.1302692. Epub 2014 Apr 11.
11. Lindsey W Plasschaert, *et al.* A single-cell atlas of the airway epithelium reveals the CFTR-rich pulmonary ionocyte. *Nature.* 2018 Aug;560(7718):377-381. doi: 10.1038/s41586-018-0394-6. Epub 2018 Aug 1.
12. Neal G Ravindra, *et al.* Single-cell longitudinal analysis of SARS-CoV-2 infection in human airway epithelium identifies target cells, alterations in gene expression, and cell state changes. *PLoS Biol.* 2021 Mar 17;19(3):e3001143. doi: 10.1371/journal.pbio.3001143. eCollection 2021 Mar.
13. Alekh Paranjapye, *et al.* Cell function and identity revealed by comparative scRNA-seq analysis in human nasal, bronchial and epididymis epithelia, *Eur J Cell Biol.* 2022 Jun-Aug;101(3):151231. doi: 10.1016/j.ejcb.2022.151231. Epub 2022 May 18.

14. Xiaoyuan Bai, *et al.* Induction of cyclophilin A by influenza A virus infection facilitates group A Streptococcus coinfection. *Cell Rep.* 2021 May 18;35(7):109159. doi: 10.1016/j.celrep.2021.109159

REVIEWER COMMENTS

Reviewer #1 (Remarks to the Author):

The authors have gone to great lengths to address most of my concerns adequately. There are two requests from my side before I deem the revised m/s acceptable:

1. The authors have now clarified how weight loss, a weight cut-off of 70% of starting weight, and mortality are linked, and this makes sense now in most figures in the m/s. To further clarify this, could the authors a. indicate whether or not weight (or mice found dead) was the only inclusion criterion for declaring a mouse dead, or whether clinical scores were also taken and mice culled when reaching a specific clinical score / clinical end point. If clinical scores were taken then this data should be included in the mouse studies.

b, along the same lines, the one figure where weight and mortality do not coincide clearly is figure 6a/b, where 100% mortality is reached without the weight mean approaching the 70% mark - please clarify and/or amend the figure.

2 In their reply 1-7, the authors show that indeed, the overlap between virus-infected cells and CD47-expressing cells is rather low, i.e. more uninfected cells express CD47 than infected. This data should be included in the m/s and should be discussed in the light of response 1-6, where they show that CD47 induction is IFN-dependent. As virus-induced IFN will be secreted and signalling on neighbouring cells, this could easily explain the expression pattern shown in response 1-7. It might be speculated that virus-infected cells have a more suppressed IFN response due to viral mechanisms suppressing IFN signals, while uninfected cells have a full response to IFN and hence up regulate CD47

Reviewer #2 (Remarks to the Author):

The authors have thoroughly addressed y two major suggestions by conducting new studies. The resulting data has made an interesting paper even stronger.

Reviewer #3 (Remarks to the Author):

The authors have done a large amount of work to try to address major points brought up by the reviewers. They use 3 different cell types (including a squamous epithelial carcinoma cell line but not polarized human bronchial epithelial cell lines – widely available); 2 mouse strains (as well as two different types of floxed mice and controls) to demonstrate that *S. aureus* binding to CD47 on some epithelial cells is an important first step in superinfection following influenza. Unfortunately, their initial lack of attention to critical details and continued overinterpretation of their results detracts from the overall impact of the work. In brief they convincingly show that *S.aureus* interacts with CD47 and this interaction contributes significantly to mortality in a murine model of superinfection. Much less clear is the actual mechanism: ? Is mortality just due to the increased bacterial load in the mouse – which need CD47 to attach? The authors show that in human nasal cells (a site of colonization but not infection) – CD47 is transiently expressed at 1 day; but in the mouse isn't induced until 5 days. Human bronchial epithelial cells (fig 1) transiently express CD47 at day 1- post flu. Moreover, there is a disconnect in the co-localization of the CD47+ ciliated cells and the ciliated cells that are infected with flu – which is mentioned but not quantified. They also confirm published data that myeloid cells need to express CD47 to clear *S. aureus*.

It is difficult to follow all of their findings, given the switching of cell types and mouse models. Why wasn't a more consistent experimental approach used? This is a concern.

Some of the more important data presented, such as the induction of CD47 in human bronchial epithelial cells with flu (Supp 8F) is simply not convincing. The mouse data – (ignoring the + but supposedly negative control) is better.

The mechanism proposed – that there is a change in the integrity of the tight junctions is not well supported. What is shown is that overwhelming bacterial infection, which they do show requires CD47, destroys mouse lung tissue accompanied by bacteremia. No specific data showing changes in the expression or function of tight junction proteins at lower (non lethal) levels of infection are provided. Nasal epithelial cells (Supp fig 2) disruption is presented as a percentage determined by histopathology – not a rigorous measurement. Cell death is not clearly quantified with LDH or LiveDead for example) but instead by showing “damaged areas” by light microscopy which is not specific. The required experiment, showing bacterial invasion was not done as was requested. This is a standard method and should not be “technically challenging” as suggested. Other data provided such as the proteomics, for example, are not relevant to the focus of this manuscript. Overall, the authors’ major contribution is demonstrating the interaction of CD47 and *S. aureus* FnBPA/B – and likely a role in nasal colonization in the setting of influenza which then facilitates *S. aureus* infection, at least in the mouse.

Reviewer #4 (Remarks to the Author):

The authors have revised the manuscript carefully. The revised manuscript can accurately and specifically display a novel role for influenza-induced epithelial CD47 in promoting attachment of *S. aureus* in vitro and in mediating severe super-infection disease in vivo. This work is of significance to the field of viral-bacterial co-infection. Based on the established literature, this work furtherly explored the mechanism of the super-infection from the new prospect. In addition, the work can support the conclusions and claims, which no additional evidence is needed. Besides, no flaw is found in the data analysis, interpretation and conclusions, which can allow the publication. In terms of methodology, it is sound and meets the expected standards in your field. And the authors provide the enough detail in the methods for the work that can be reproduced. Above all, this paper meets the requirements of the publication and can be allowed to be published.

Reviewer #1 (Remarks to the Author):

1. The authors have now clarified how weight loss, a weight cut-off of 70% of starting weight, and mortality are linked, and this makes sense now in most figures in the m/s. To further clarify this, could the authors a. indicate whether or not weight (or mice found dead) was the only inclusion criterion for declaring a mouse dead, or whether clinical scores were also taken and mice culled when reaching a specific clinical score / clinical end point. If clinical scores were taken then this data should be included in the mouse studies. b, along the same lines, the one figure where weight and mortality do not coincide clearly is figure 6a/b, where 100% mortality is reached without the weight mean approaching the 70% mark - please clarify and/or amend the figure.

[Response 1-1a] We appreciate your thorough review and thoughtful comments on our revised manuscript. Regarding your inquiry into the inclusion criteria for declaring a mouse dead, we would like to clarify that weight loss (or discovery of mice found dead) was the sole criterion for determining mortality in our study. Clinical scores were not used as an additional parameter for declaring a mouse dead. This clarification has been incorporated into the Methods section of the revised manuscript (lines 575-577).

[Response 1-1b] Regarding Figure 6a/b, we confirm that all mice in the *CD47^{fl/fl}* + FnBP A+B+ group were found dead before reaching the 70% weight loss threshold. Specifically, seven mice were found dead on day 7 and two mice on day 8. As a result, we declared these mice dead based on the sole criterion mentioned above.

2 In their reply 1-7, the authors show that indeed, the overlap between virus-infected cells and CD47-expressing cells is rather low, i.e. more uninfected cells express CD47 than infected. This data should be included in the m/s and should be discussed in the light of response 1-6, where they show that CD47 induction is IFN-dependent. As virus-induced IFN will be secreted and signalling on neighbouring cells, this could easily explain the expression pattern shown in response 1-7. It might be speculated that virus-infected cells have a more suppressed IFN response due to viral mechanisms suppressing IFN signals, while uninfected cells have a full response to IFN and hence up regulate CD47

[Response 1-2] Thank you for your insightful comment. As requested, we have included the relevant data as Supplementary Figure 3g. Additionally, we have integrated the following sentence into the Discussion section (lines 396-403) to address the relationship between virus-infected cells, CD47 expression, and the IFN-dependent induction: "It's important to note that while CD47 induction by viral infection is specific to ciliated cells (Fig. 1e and Supplementary Fig. 2d), not all ciliated cells were infected (**revised Supplementary Fig. 3g**). Given that virus-induced IFNs are secreted and signal on neighboring cells, the observed expression pattern can be explained. Although it could be speculated that virus-infected cells experience a more suppressed IFN response due to viral mechanisms inhibiting IFN signals, whereas uninfected cells exhibit a full response to IFN, resulting in the upregulation of CD47, the precise mechanism by which neighboring infected cells induce CD47 in ciliated cells remains unclear at this stage."

We believe these revisions adequately address your concerns and provide a more comprehensive explanation of our experimental procedures. Thank you for guiding us toward enhancing the presentation of our research. We appreciate the opportunity to clarify and improve our manuscript.

Reviewer #2 (Remarks to the Author):

The authors have thoroughly addressed y two major suggestions by conducting new studies. The resulting data has made an interesting paper even stronger.

[Response] Thank you for your thoughtful review and valuable feedback. We are pleased to hear that our efforts to address your two major suggestions have been successful and that the additional studies have strengthened the overall quality of the paper.

Reviewer #3 (Remarks to the Author):

[3-1] The authors have done a large amount of work to try to address major points brought up by the reviewers. They use 3 different cell types (including a squamous epithelial carcinoma cell line but not polarized human bronchial epithelial cell lines – widely available); 2 mouse strains (as well as two different types of floxed mice and controls) to demonstrate that *S. aureus* binding to CD47 on some epithelial cells is an important first step in superinfection following influenza. Unfortunately, their initial lack of attention to critical details and continued overinterpretation of their results detracts from the overall impact of the work. In brief they convincingly show that *S. aureus* interacts with CD47 and this interaction contributes significantly to mortality in a murine model of superinfection.

[Response 3-1] To address the reviewer’s concern regarding the critical details and potential overinterpretation of our results, we have conducted additional experiments, including bacterial invasion assays and Western blot analysis for ZO-1 dysregulation using our *in vivo* mouse model. We have demonstrated that bacterial invasion was increased during super-infection and significantly inhibited by ciliated cell-specific CD47 gene deletion, FnBP mutation in *S. aureus*, or treatment with anti-CD47 neutralizing antibodies (**revised Supplementary Figure 10g, revised Figure 4f, p, revised Figure 5f, and revised Figure 6f**). Moreover, we have validated the downregulation of ZO-1 in the lung following viral infection. Consequently, we have included a comprehensive discussion of our research (lines 434-443): “Due to the methodological challenges inherent in our primary epithelial cell ALI culture setup, we were unable to distinctly differentiate between bacterial adherence and invasion *in vitro*. Instead, we utilized a super-infection mouse model to evaluate both the adherence of *S. aureus* and its invasion into the lung tissue, as well as bacterial CFU in the spleen as an indicator of systemic bacterial dissemination. In order to establish an *in vivo* correlation with the observed loss of the epithelial barrier function in our *in vitro* super-infection model (Supplementary Fig. 5 and Fig. 2), we validated the downregulation of ZO-1 in the lung following viral infection (**revised Supplementary Fig. 8f**) and the increases in *S. aureus* invasion in the lung (**revised Supplementary Fig. 10g and Fig. 5f**) and bacteremia during super-infection (Supplementary Fig. 10h and Fig. 5g).” Although we have provided additional data to strengthen our mechanistic explanations regarding epithelial barrier disruption, we acknowledge the limitations of our study as outlined in the Discussion section: “Nonetheless, we have not yet elucidate the precise mechanisms by which accumulated bacteria breach the barrier more effectively. Further investigation into how *S. aureus* interacts with CD47 to facilitate its invasion process remains a valuable and ongoing area of research (lines 448-451)”.

These revisions aim to address the concerns raised by the reviewer and provide a more balanced interpretation of our findings.

[3-2] Much less clear is the actual mechanism: ? Is mortality just due to the increased bacterial load in the mouse – which need CD47 to attach? The authors show that in human nasal cells (a site of colonization but not infection) – CD47 is transiently expressed at 1 day; but in the mouse isn’t induced until 5 days. Human

bronchial epithelial cells (fig 1) transiently express CD47 at day 1- post flu.

[Response 3-2] In addressing the observed difference in CD47 expression between human primary epithelial cells and mouse lung, it's essential to consider potential temporal disparities inherent to *in vitro* culture systems versus *in vivo* mouse models. Unlike *in vitro* experiments where the virus is directly introduced to the culture media, in our *in vivo* model, mice were infected nasally, and we subsequently analyzed their lung tissue for CD47 induction. This route of infection may lead to a delayed onset of CD47 expression in the lung compared to direct infection *in vitro*. For example, it may take several days for nasally infected virus to reach the lung and induce CD47 expression. This distinction in experimental methodology likely accounts for the observed differences in CD47 expression kinetics between human nasal cells and mouse lung tissues.

[3-3] Moreover, there is a disconnect in the co-localization of the CD47+ ciliated cells and the ciliated cells that are infected with flu – which is mentioned but not quantified.

[Response 3-3] Thank you for your valuable feedback. In response, we have addressed this concern by providing quantified data regarding the co-localization of CD47+ ciliated cells and the ciliated cells infected with influenza in **revised Supplementary Figure 3g**. Reviewer #1 also raised a similar concern, and we have included the relevant data as Supplementary Figure 3g. Additionally, we have integrated the following sentence into the Discussion section (lines 396-403) to address the relationship between virus-infected cells, CD47 expression, and the IFN-dependent induction: "It's important to note that while CD47 induction by viral infection is specific to ciliated cells (Fig. 1e and Supplementary Fig. 2d), not all ciliated cells were infected (revised Supplementary Fig. 3g). Given that virus-induced IFNs are secreted and signal on neighboring cells, the observed expression pattern can be explained. Although it could be speculated that virus-infected cells experience a more suppressed IFN response due to viral mechanisms inhibiting IFN signals, whereas uninfected cells exhibit a full response to IFN, resulting in the upregulation of CD47, the precise mechanism by which neighboring infected cells induce CD47 in ciliated cells remains unclear at this stage."

[3-4] They also confirm published data that myeloid cells need to express CD47 to clear *S. aureus*.

[Response 3-4] As the reviewer noted, previous studies (Nauseef 2014 JI) have highlighted the crucial role of CD47 as an "eat me" signal for macrophage clearance of neutrophils containing *S. aureus*, leading to increased infection in the absence of CD47. We acknowledge the importance of myeloid cell expression of CD47 in clearing *S. aureus*. In our manuscript, we emphasize a novel aspect where CD47 expressed in epithelial cells during viral infection serves as an attachment site for secondary bacterial infection. While previous research has focused on CD47's role in immune cell communication during bacterial infection, our findings suggest a distinct function for CD47 in epithelial cells. We have added the following sentence into the Discussion section (lines 459-466): "A previous study has highlighted the crucial role of CD47 as an "eat me" signal critical for macrophage clearance of neutrophils that have ingested *S. aureus*, leading to increased infection in the absence of CD47. While prior research has focused on CD47's involvement in immune cell communication during bacterial infection, our findings emphasize a novel aspect where CD47 expressed in epithelial cells during viral infection serves as an attachment site for secondary bacterial infection. Hence, it is conceivable that the roles of CD47 on epithelial cells versus myeloid cells are distinct, necessitating further investigation to fully understand these differences."

[3-5] It is difficult to follow all of their findings, given the switching of cell types and mouse models. Why wasn't a more consistent experimental approach used? This is a concern.

[Response 3-5] We appreciate the reviewer for raising this question. While we acknowledge that our experimental approach may appear inconsistent and difficult to follow, we initially needed to utilize a cell

culture system as an *in vitro* model and later transitioned to a mouse model as an *in vivo* model to investigate the role of CD47 during super-infection. This approach allowed us to explore different aspects of the interaction between *S. aureus* and CD47 in both cellular and whole-organism contexts. To address consistency concerns regarding cell types, during the first revision process, we replaced most of the data using the A549 cell line with data using HBECs. In the case of the mouse model, we employed *CD47^{fl}*, *CD47^{Foxj1}*, and *CD47^{LysM}* mice for the CD47 gene deletion experiment and C57BL/6 WT mice for the CD47 neutralizing experiment. Consequently, we had to conduct control experiments using two distinct sets of mice to evaluate the impact of our experimental interventions. While complete consistency couldn't be achieved due to the nature of the study, we made our best efforts in this regard.

[3-6] Some of the more important data presented, such as the induction of CD47 in human bronchial epithelial cells with flu (Supp 8F) is simply not convincing. The mouse data – (ignoring the + but supposedly negative control) is better.

[Response 3-6] Thank you for bringing this concern to our attention. In response, we have performed Western blotting assays and replaced the previously presented data with a new blot, now depicted as **revised Supplementary Figure 8g and Supplementary Figure 8h**. We believe this updated data provides a more convincing representation of the induction of CD47 in HBECs and the mouse lung in response to influenza virus infection.

[3-7] The mechanism proposed – that there is a change in the integrity of the tight junctions is not well supported. What is shown is that overwhelming bacterial infection, which they do show requires CD47, destroys mouse lung tissue accompanied by bacteremia. No specific data showing changes in the expression or function of tight junction proteins at lower (non lethal) levels of infection are provided.

[Response 3-7] In order to provide specific data supporting the proposed mechanism and to demonstrate changes in the integrity of tight junctions at lower (non-lethal) levels of infection, we have evaluated the expression of ZO-1 in the lungs of mice infected solely with the virus at 5 dpi. This analysis is depicted in **revised Supplementary Figure 8f**, validating the downregulation of ZO-1 in the lung following viral infection.

[3-8] Nasal epithelial cells (Supp fig 2) disruption is presented as a percentage determined by histopathology – not a rigorous measurement. Cell death is not clearly quantified with LDH or LiveDead for example) but instead by showing “damaged areas” by light microscopy which is not specific.

[Response 3-8] The disruption of barrier integrity in airway epithelial cells was presented as a fluorescence intensity determined by paracellular permeability of fluorescein isothiocyanate (FITC)-dextran or trans-epithelial electrical resistance (TEER) at 3 dpi in both HBECs (Figure 2) and HNECs (Supplementary Figure 5). The LDH assay was employed to ensure that this disruption was not merely attributed to cell death, demonstrating the relative quantification of cell death by comparing positive and negative controls. The identification of damaged areas via light microscopy was utilized to illustrate the consequences of super-infection at 7 dpi for HNECs and 5 dpi for HBECs.

[3-9] The required experiment, showing bacterial invasion was not done as was requested. This is a standard method and should not be “technically challenging” as suggested.

[Response 3-9] In response to the reviewer's request, we have performed bacterial invasion assays in our mouse super-infection model, as demonstrated in **revised Supplementary Figure 10g, revised Figure 4f, p, revised Figure 5f, and revised Figure 6f**. We have observed that bacterial invasion increased during super-infection was inhibited by ciliated cell-specific CD47 gene deletion, FnBP mutation in *S. aureus*, or

treatment of anti-CD47 neutralizing antibodies. While we acknowledge the importance of conducting similar assays in HBECs, we encountered technical obstacles in doing so, which we have duly noted. As far as we are aware, there has been no prior study quantifying bacterial invasion following secondary bacterial infection using HBECs.

We appreciate the reviewer's feedback and have endeavored to address their concerns by providing additional data and clarifications.

[3-10] Other data provided such as the proteomics, for example, are not relevant to the focus of this manuscript.

Overall, the authors' major contribution is demonstrating the interaction of CD47 and *S. aureus* FnBPA/B – and likely a role in nasal colonization in the setting of influenza which then facilitates *S. aureus* infection, at least in the mouse.

[Response 3-10] Thank you for the reviewer's feedback. Although we understand that the proteomics data may seem unrelated to the main focus of the manuscript, it actually played a crucial role in identifying the target protein CD47 at the outset of this study. Consequently, we believe it is important to retain this data, albeit as supplementary material (Supplementary Figure 1), to provide context and insight into the study's inception.

Reviewer #4 (Remarks to the Author):

The authors have revised the manuscript carefully. The revised manuscript can accurately and specifically display a novel role for influenza-induced epithelial CD47 in promoting attachment of *S. aureus* in vitro and in mediating severe super-infection disease in vivo. This work is of significance to the field of viral-bacterial co-infection. Based on the established literature, this work furtherly explored the mechanism of the super-infection from the new prospect. In addition, the work can support the conclusions and claims, which no additional evidence is needed. Besides, no flaw is found in the data analysis, interpretation and conclusions, which can allow the publication. In terms of methodology, it is sound and meets the expected standards in your field. And the authors provide the enough detail in the methods for the work that can be reproduced. Above all, this paper meets the requirements of the publication and can be allowed to be published.

[Response] We sincerely thank you for your time, expertise, and positive assessment. Your support is invaluable, and we are grateful for the opportunity to contribute to the scientific community.

REVIEWERS' COMMENTS

Reviewer #3 (Remarks to the Author):

The authors have made a considerable effort to improve the manuscript with new data in both the supplementary and manuscript figures that help to support their central hypothesis. Importantly, the discussion now includes the major points raised regarding the co-localization of CD47 and influenza infection as well as the distribution of CD47 in nasal versus bronchial cells in the various models used. This is much improved.

Point-By-Point Response to Reviewers' Comments

Reviewer #3 (Remarks to the Author):

The authors have made a considerable effort to improve the manuscript with new data in both the supplementary and manuscript figures that help to support their central hypothesis. Importantly, the discussion now includes the major points raised regarding the co-localization of cd 47 and influenza infection as well as the distribution of CD47 in nasal versus bronchial cells in the various models used. This is much improved.

[Response] Thank you very much for your valuable feedback. We sincerely appreciate your recognition of the efforts made to enhance the manuscript, including the addition of new data in both the supplementary and manuscript figures. We are pleased that the revisions have strengthened the support for our central hypothesis. Additionally, we are glad that the discussion now adequately addresses the major points raised regarding the co-localization of CD47 and influenza infection, as well as the distribution of CD47 in nasal versus bronchial cells in the various models used. Your comments have been instrumental in improving the quality and clarity of our work.